# SocialJax: An Evaluation Suite for Multi-agent Reinforcement Learning in Sequential Social Dilemmas

**Zihao Guo**[1][*] **Shuqing Shi**[1][*] **Richard Willis**[1] **Tristan Tomilin**[2] **Joel Z. Leibo**[4] **Yali Du**[1,3]

[1]King's College London, [2]Eindhoven University of Technology, [3]The Alan Turing Institute, [4]Google DeepMind

{zihao.1.guo, shuqing.shi, richard.willis, yali.du}@kcl.ac.uk
t.tomilin@tue.nl, jzl@google.com

## Abstract

Sequential social dilemmas pose a significant challenge in the field of multi-agent reinforcement learning (MARL), requiring environments that accurately reflect the tension between individual and collective interests. Previous benchmarks and environments, such as Melting Pot, provide an evaluation protocol that measures generalization to new social partners in various test scenarios. However, running reinforcement learning algorithms in traditional environments requires substantial computational resources. In this paper, we introduce SocialJax, a suite of sequential social dilemma environments and algorithms implemented in JAX. JAX is a high-performance numerical computing library for Python that enables significant improvements in operational efficiency. Our experiments demonstrate that the SocialJax training pipeline achieves at least 50× speed-up in real-time performance compared to Melting Pot's RLlib baselines. Additionally, we validate the effectiveness of baseline algorithms within SocialJax environments. Finally, we use Schelling diagrams to verify the social dilemma properties of these environments, ensuring that they accurately capture the dynamics of social dilemmas. Our code is available at https://github.com/cooperativex/SocialJax.

## 1 Introduction

Solving sequential social dilemmas remains a pivotal challenge in multi-agent reinforcement learning (MARL). Efficient and rich environments and benchmarks are crucial for enabling rigorous evaluation and meaningful comparison of algorithms. Traditional MARL environments (Lowe et al., 2017; Samvelyan et al., 2019; Bard et al., 2020; Carroll et al., 2019) have overwhelmingly focused on fully cooperative or competitive tasks. Only a handful of benchmarks, such as Prisoner's Dilemma of OpenSpiel (Lanctot et al., 2019), Stag Hunt and Chicken in PettingZoo (Terry et al., 2021), address social dilemmas. However, these environments rely on CPU-based parallelism, which limits their simulation throughput and scalability, and they offer only a narrow variety of scenarios without standardized evaluation protocols for sequential social dilemmas.

Although there are environments (Leibo et al., 2021; Agapiou et al., 2022) specifically designed for sequential social dilemmas, conducting experiments in these environments is highly challenging due to the substantial number of environment time steps required for training. Training in such environments demands extremely high hardware specifications (over 1000 CPUs for Melting Pot) or a significant amount of computing time. Moreover, while the Melting Pot framework (Agapiou et al., 2022; Leibo et al., 2021) includes several baseline algorithms, such as A3C (Mnih et al., 2016), V-MPO (Song et al., 2020), and OPRE (Vezhnevets et al., 2020), these implementations are not publicly available, which poses challenges for reproducibility and further development.

Recent libraries such as JAX (Bradbury et al., 2018) facilitate efficient execution of NumPy-style Python code on hardware accelerators(GPUs and TPUs) by automatically translating high-level

---

[*]Equal contribution.

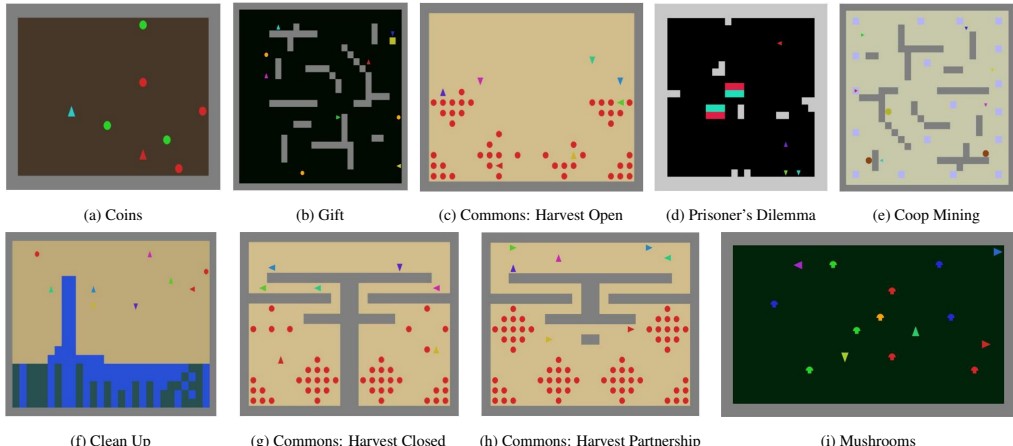

Figure 1: The SocialJax suite contains nine multi-agent reinforcement learning environments designed to evaluate social dilemmas. The agents of all environments are restricted to partial observability of their surroundings through a designated observation window.

functions into optimized, parallelizable operations. In reinforcement learning(RL) research, several approaches like PureJaxRL (Lu et al., 2022) have been proposed to leverage JAX (Bradbury et al., 2018) to run environments on GPUs. Parallelizing environments on GPUs could significantly accelerate computation. However, PureJAXRL is limited to single-agent environments and does not support multi-agent scenarios. JaxMARL (Rutherford et al., 2024) extends JAX's efficiency gains to MARL by providing a suite of JAX-based environments and algorithms. While JaxMARL incorporates a limited set of social dilemma environments (e.g., Coins and STORM), it remains primarily geared toward traditional cooperative tasks and lacks dedicated benchmarks specifically designed for sequential social dilemmas.

In this paper, we develop a suite of environments for sequential social dilemmas based on JAX to improve computational efficiency. We also provide JAX-based implementations of multiple MARL algorithms and construct a complete training pipeline, significantly accelerating the training process. Solving sequential social dilemma problems often requires a large number of time steps and diverse evaluation environments. SocialJax addresses these challenges by offering both computationally efficient and diverse environments. Furthermore, we validate the social dilemma properties of each environment using Schelling diagrams. In addition, we design metrics beyond return for each environment, enabling a more precise quantification of the cooperative and competitive behaviors of agents. To the best of our knowledge, our work is the first evaluation framework leveraging JAX for sequential social dilemmas. Our contributions are as follows:

**Implementations of Social Dilemma Environments**: We implement a diverse suite of sequential social dilemma environments in JAX (see Figure 1), each exhibiting mixed-incentive dynamics, and achieve substantial improvements in simulation speed.

**Implementation of Algorithms in JAX**: We implement a series of JAX-based MARL algorithms, including Independent PPO (IPPO) (De Witt et al., 2020; Rutherford et al., 2024) under both individual and common reward settings, Multi-Agent PPO (MAPPO) (Yu et al., 2022), Value Decomposition Networks (VDN) (Sunehag et al., 2018), Social Value Orientation (SVO) (McKee et al., 2020), and PPO using reward exchange (IPPO-RE) (Willis et al., 2025). We also provide training results with and without parameter sharing. We show that training algorithms in SocialJax is at least 50 times faster than Melting Pot 2.0 on a machine with 14 CPUs and 1 Nvidia A100 GPU.

**Performance Benchmark and Analysis**: We benchmark the performance of various environments and algorithms in terms of simulation throughput and training speed. Additionally, we evaluate the performance of the algorithms we include across various environments. Finally, we design specific evaluation metrics for each environment to measure agent behaviors as either prosocial or antisocial. We further analyze the behavior of selfish and cooperative agents to understand the underlying incentive structures using Schelling diagrams (Pérolat et al., 2017; Schelling, 1973). Finally, we analyze the performance of the implemented algorithms in our environments, including cases where agent behavior varies with different SVO (McKee et al., 2020) reward angles.

## 2 RELATED WORK

In this section, we review related work on sequential social dilemmas (Du et al., 2023) and JAX-based reinforcement learning environments, algorithms, and benchmarks. Sequential Social Dilemma (Vinitsky et al., 2019) introduces a testbed for evaluating social dilemmas. However, its scope is restricted to only two environments on Clean Up and Harvest. OpenSpiel (Lanctot et al., 2019) offers a collection of environments, some of which feature both cooperative and competitive dynamics, allowing for the exploration of social dilemmas. Melting Pot (Leibo et al., 2021) was the first to establish a standardized benchmark for MARL specifically designed to evaluate agent performance in sequential social dilemmas. By providing a diverse set of carefully designed multi-agent scenarios, it enables systematic testing of agent behaviors in cooperative, competitive, and mixed-motive settings. Melting Pot 2.0 (Agapiou et al., 2022) introduces a more comprehensive and refined suite of testing environments that capture a wide range of interdependent relationships and incentive structures. The existing social dilemma environments are CPU-based, and their execution efficiency is constrained by CPU performance. Recent work also explores mechanism design approaches to social dilemmas (Zhang et al., 2025).

JAX (Bradbury et al., 2018) is an open-source library developed by Google, designed for high-performance numerical computing and machine learning tasks. Based on the GPU and TPU acceleration capabilities introduced by JAX, along with its support for parallelism and vectorization, JAX can effectively accelerate both environments and algorithms on hardware accelerators (GPUs, TPUs). Functorch (He & Zou, 2021) provides JAX-like composable function transforms for PyTorch, making it easier to accelerate RL algorithms on hardware accelerators. (Lu et al., 2022) introduces a parallelized framework leveraging JAX, enabling both environment and model training to run on GPUs. Jumanji (Bonnet et al., 2024) has developed a diverse range of single-agent environments in JAX, ranging from simple games to NP-hard combinatorial problems. Gymnax (Lange, 2022) provides JAX versions of different environments, including classic control, Bsuite (Osband et al., 2020), MinAtar (Young & Tian, 2019), and a collection of classic RL tasks. Pgx (Koyamada et al., 2023) implements a variety of board game simulation environments using JAX, such as Chess, Shogi, and Go. Brax (Freeman et al., 2021) is a physics engine written in JAX for RL, which re-implements MuJoCo. XLand-MiniGrid (Nikulin et al., 2024) is a grid-world environment and benchmark fully implemented in JAX. Similarly, our work is also demonstrated using grid-world environments. Mava (Mahjoub et al., 2024) and JaxMARL (Rutherford et al., 2024) both provide JAX-based MARL environments and baseline algorithms. Among the JAX environments and algorithms, only the Storm and Coins environments in JaxMARL are related to social dilemmas, while none of them specifically address sequential social dilemmas. We construct multiple environments focused on social dilemma challenges.

## 3 SOCIALJAX

In this section, we define the concept of social dilemmas, introduce the environments we implement, describe the algorithms used, and present the metrics we design to measure cooperative and competitive behaviors. Our work presents an efficient testing framework for sequential social dilemmas, complete with evaluation environments and benchmark baselines. Our SocialJax encompasses a diverse suite of environments, which collectively capture a wide range of sequential social dilemma scenarios, including public good dilemmas and common pool resource problems. We evaluate the environments using IPPO individual reward (both with and without parameter sharing), IPPO common reward, MAPPO, SVO and IPPO-RE.

### 3.1 SEQUENTIAL SOCIAL DILEMMAS

In $N$-player partially observable Markov games (Littman, 1994), agents possess partial observation of the environment. At each state, players choose actions from their respective action sets $\mathcal{A}^1, \ldots, \mathcal{A}^N$. State transitions $\mathcal{T} : \mathcal{S} \times \mathcal{A}^1 \times \ldots \times \mathcal{A}^N \to \Delta(\mathcal{S})$ are determined by the current state and the joint actions taken by all agents, where $\Delta(\mathcal{S})$ represents the set of discrete probability distributions over $\mathcal{S}$. We define the observation space of player $i$ as $O_i = \{o^i | s \in \mathcal{S}, o^i = O(s, i)\}$. Each agent has its corresponding individual reward defined as $r^i : \mathcal{S} \times \mathcal{A}^1 \times \cdots \times \mathcal{A}^N \to \mathbb{R}$. In an $N$-player *sequential social dilemma*, cooperators and defectors each have their own disjoint

policies $(\pi_c^1, \ldots, \pi_c^l, \ldots, \pi_d^1, \ldots, \pi_d^m) \in \Pi_c^l \times \Pi_d^m$ with $l + m = N$. $l$ is the number of cooperators. $R_c(l)$ and $R_d(l)$ are denoted as the average payoff for cooperating policies and defecting policies. When we have a sequential social dilemma, if it satisfies the following conditions (Leibo et al., 2017; Hughes et al., 2018):

1. Mutual cooperation is greater than mutual defection: $R_c(N) > R_d(0)$.

2. Mutual cooperation is greater than being exploited by defectors: $R_c(N) > R_c(0)$.

3. Either the fear property or the greed property (or both) holds:
   Fear: $R_d(i) > R_c(i)$ for sufficiently small $i$;    Greed: $R_d(i) > R_c(i)$ for sufficiently large $i$.

We use Schelling diagrams (Pérolat et al., 2017; Schelling, 1973) to depict the payoffs of different numbers of cooperators and defectors. The lines $R_c(l + 1)$ and $R_d(l)$ depict the average reward for an agent choosing to either cooperate or defect, as a function of the number of co-players that are cooperating (see Figure 3).

## 3.2 Environments of SocialJax

Figure 1 shows the layout of our environments. SocialJax environments are mainly derived from Melting Pot 2.0 (Agapiou et al., 2022). However, the layout of SocialJax environments is not the same as Melting Pot, but rather more similar to MiniGrid (Chevalier-Boisvert et al., 2018). Agents observe a grid matrix, with different objects represented by distinct numerical values. In all our environments, agents have the same $11 \times 11$ field of view implemented at the grid level. For further details, please refer to the Appendix B. Next, we detail the environment setup, including the specific environments in our evaluation suite.

**Coins** This environment was first introduced by (Lerer & Peysakhovich, 2017) to investigate how agents behave in situations where they can cooperate with or harm others by eating other agent's type of coin, making it a classic example for social dilemma.

**Commons Harvest** Agents need to consume apples to obtain rewards; however, the probability of apple regrowth is determined by the number of apples in the local neighborhood. This creates tension between selfish agents and cooperative agents. In order to discuss different social behaviors of agents, in Commons Harvest, we established three different scenarios: *Commons Harvest: Closed*, *Commons Harvest: Open*, and *Commons Harvest: Partnership*.

In *Commons Harvest: Closed*, there are two rooms in this setup (Pérolat et al., 2017), each containing multiple patches of apples and a single entry. These rooms can be defended by specific players to prevent others from accessing the apples by zapping them with a beam, causing them to respawn.

In *Commons Harvest: Open*, all individuals can harvest apples, but must restrict their actions to not collect the last apple and ensure that the apples can regrow.

The *Commons Harvest: Partnership* is similar to Commons Harvest: Closed, specific players are required to defend the room. However, in this case, each room has two entry points, necessitating two players to cooperate in defending the room.

**Clean Up** This environment is a public good game where players earn rewards by collecting apples. But the spawn rate of apples depends on the cleanliness of a nearby river. For continuous apple growth, the agents must keep river pollution levels consistently low over time.

**Coop Mining** In this environment, two types of ore spawn randomly in empty spaces. Iron ore (gray) can be mined individually and provides a reward of +1 upon extraction, whereas gold ore requires group coordination and yields a higher payoff per miner. Selfish agents tend to mine the iron ore more, while cooperative agents will try to cooperate and mine gold ore.

**Mushrooms** This environment includes multiple types of mushrooms that, when consumed by agents, deliver different rewards. Some mushrooms benefit only the individual agent, while others are beneficial to the group.

**Gift Refinement** In this environment, agents can give collected tokens to other agents on the map, which upgrade to yield greater rewards. Therefore, maximizing returns requires agents to trust one another and cooperate effectively.

**Prisoner's Dilemma: Arena** Agents inside this environment collect "defect" or "cooperate" tokens and then interact with each other to compare inventories. The resulting payoffs follow the classic Prisoner's Dilemma matrix, emphasizing the tension between individual incentives and collective welfare.

### 3.3 ALGORITHMS IN JAX

Due to IPPO's widespread adoption and stability (Schulman et al., 2017; De Witt et al., 2020), we implement IPPO under both parameter-sharing and non-parameter-sharing settings. The parameter-sharing variant of IPPO leads to faster training by using a shared policy neural network across all agents. The non-parameter-sharing variant, although slower, preserves agent-specific policies that prevent convergence to uniform conventions. The IPPO training curves shown in Figure 2 are based on the parameter sharing variant. Additional comparisons between parameter sharing and non-parameter sharing IPPO can be found in the Appendix C.1. To encourage prosocial behavior, we introduce common rewards, while individual rewards are used to incentivize selfishness in agents. Additionally, we include MAPPO (Yu et al., 2022) as a representative centralized learning algorithm for MARL. Finally, we implement SVO (McKee et al., 2020) and reward exchange (Willis et al., 2025) as representative methods specifically designed to handle social dilemmas, where agents can balance individual and collective interests.

**IPPO Common Rewards** refers to a scenario where all agents in a multi-agent system share a single, unified reward signal derived from the environment. This approach ensures that all agents are aligned towards achieving the same objective, promoting collaboration and coordination among them. By sharing a common reward, agents are incentivized to work together and make decisions that benefit the collective, rather than focusing solely on individual gains. This can help prevent conflicts and encourage cooperative behavior.

**IPPO Individual Rewards** indicates that each agent is assigned its own reward, inherently encouraging selfish behavior. This approach allows agents to prioritize maximizing their personal success over collective goals.

**MAPPO** We implement Multi-Agent PPO (MAPPO) (Yu et al., 2022) using both the PureJaxRL (Lu et al., 2022) and JaxMARL (Rutherford et al., 2024) frameworks. As a centralized learning approach, MAPPO uses the centralized value function to help incentivize agents to perform prosocial behaviors.

**VDN** (Sunehag et al., 2018) decomposes the team's joint Q-value into a sum of individual agents' local Q-values, and achieves centralized training with decentralized execution by maximizing the aggregated value.

**SVO** We implement SVO (McKee et al., 2020) as an intrinsic motivation framework to encourage cooperative behavior in our environments. To capture the relative reward distribution, *reward angle* $\theta(R) = \arctan\left(\frac{\bar{r}_{-i}}{r_i}\right)$ is defined by SVO, where $r_i$ is the reward received by agent $i$, and $\bar{r}_{-i} = \frac{1}{n-1}\sum_{j\neq i} r_j$ is the average reward of all other agents. Each agent is assigned a target SVO angle $\theta^{\text{SVO}}$ that reflects its desired distribution of reward. The agent's utility function combines extrinsic and intrinsic rewards as:

$$U_i(s, o_i, a_i) = r_i - w \cdot |\theta^{\text{SVO}} - \theta(R)|,$$

where $w$ controls the strength of the Social Value Orientation. We restrict SVO preferences to the non-negative quadrant, i.e., $\theta \in [0°, 90°]$, where $\theta = 90°$ represents an Altruistic setting and $\theta = 0°$ represents an Individualistic setting.

**IPPO-RE** Reward exchange (Willis et al., 2025) is a mechanism whereby agents exchange portions of their per-timestep rewards. Each agent retains a proportion $s$ of its own reward (its *self-interest*) and distributes the remainder equally among co-players. Agent $i$'s utility under reward exchange is:

$$U_i(\bar{r}, s) = s\, r_i + \frac{1-s}{n-1}\sum_{j\neq i} r_j.$$

Under reward exchange, the ratio of an agent's utility gain when a co-player receives reward to its gain when it receives the same reward itself ranges from 0 (individual reward, $s = 1$) to 1 (common reward, $s = \frac{1}{n}$). Reward exchange aims to find a balance between these extremes, with agents

exchanging sufficient reward to induce good collective outcomes while maintaining individual reward signals. We interpolate between the individual and common reward by training IPPO at reward ratios of $\frac{1}{4}$, $\frac{1}{2}$, and $\frac{3}{4}$. For example, in Coins (a two-player environment), we use $s = \frac{4}{5}$, $s = \frac{2}{3}$, and $s = \frac{4}{7}$.

## 3.4 Metrics for SocialJax Environments

Since an agent's behavior in the environments cannot be fully captured by the return alone, we devised a tailored metric for each environment to quantify whether agents tend toward cooperation or competition. We define the semantics of the metrics for each environment as follows:

**Coins**: Since each agent collecting their own-color coin does not harm the other, whereas collecting the other agent's coin imposes a penalty on them, we evaluate cooperation by counting how many coins each agent collects that match their assigned color.

**Commons Harvest**: In this environment, agents must maintain a sustainable number of apples to ensure they continue to respawn. Therefore, we measure the number of apples remaining on the map to assess whether agents are capable of preserving the long-term collective interest.

**Clean Up**: Agents need to clean polluted river tiles to enable apple regrowth. We evaluate their cooperative or selfish behavior by counting the number of cleaned water tiles.

**Coop Mining**: In this environment, agents can obtain higher rewards by cooperating to mine gold, while those who choose not to cooperate can only mine iron for a smaller reward. Therefore, we evaluate agents' cooperative behavior by measuring the amount of gold mined.

**Mushrooms**: The environment contains multiple types of mushrooms. Consuming blue mushrooms does not benefit the agent itself but provides rewards to the other agents. We thus measure the number of blue mushrooms consumed to evaluate the agent's willingness to sacrifice for the common good.

**Gift Refinement**: In this environment, agents can choose to gift collected tokens to others, which results in receiving a larger reward in return. We thus count the number of received tokens to evaluate whether agents are willing to trust each other.

**Prisoner's Dilemma: Arena**: The environment contains two types of resources, cooperative and competitive, and we introduce the quantity of cooperative resources collected by agents as a metric.

## 4 Experiments

### 4.1 Setup

To validate our library, we perform thorough tests of Common Reward IPPO, Individual Reward IPPO, MAPPO, and SVO algorithms across a diverse range of Sequential Social Dilemma environments and analyze the returns and designed cooperation metrics. We assess the performance of our JAX-based environment on a GPU and compare its speed with the implementations provided by Melting Pot 2.0 (Agapiou et al., 2022). In addition to these experiments, we plot Schelling diagrams to further verify that our environments effectively represent social dilemmas.

Table 1: Results on environment steps per second across various SocialJax environments under random actions. Different environment configurations are tested at scales from 1 to 4096 JAX Env.

| Envs | 1 Original Env | 1 JAX Env | 128 JAX Env | 1024 JAX Env | 4096 JAX Env |
|------|------|------|------|------|------|
| Coins | $1.2 \times 10^4$ | $2.0 \times 10^3$ | $2.6 \times 10^5$ | $1.4 \times 10^6$ | $3.4 \times 10^6$ |
| Harvest: Open | $3.7 \times 10^3$ | $1.2 \times 10^3$ | $1.1 \times 10^5$ | $5.0 \times 10^5$ | $7.9 \times 10^5$ |
| Harvest: Closed | $3.7 \times 10^3$ | $1.2 \times 10^3$ | $1.1 \times 10^5$ | $5.0 \times 10^5$ | $7.9 \times 10^5$ |
| Harvest: Partnership | $3.7 \times 10^3$ | $1.2 \times 10^3$ | $1.1 \times 10^5$ | $5.0 \times 10^5$ | $7.9 \times 10^5$ |
| Clean Up | $2.7 \times 10^3$ | $1.1 \times 10^3$ | $1.0 \times 10^5$ | $4.3 \times 10^5$ | $6.1 \times 10^5$ |
| Coop Mining | $3.6 \times 10^3$ | $1.9 \times 10^3$ | $1.9 \times 10^5$ | $1.0 \times 10^6$ | $1.5 \times 10^6$ |
| Mushrooms | $4.2 \times 10^3$ | $1.4 \times 10^3$ | $1.4 \times 10^5$ | $6.3 \times 10^5$ | $9.8 \times 10^5$ |
| Gift Refinement | $4.1 \times 10^3$ | $1.5 \times 10^3$ | $1.5 \times 10^5$ | $7.0 \times 10^5$ | $1.2 \times 10^6$ |
| Prisoner's Dilemma: Arena | $4.5 \times 10^3$ | $2.2 \times 10^3$ | $2.3 \times 10^5$ | $1.3 \times 10^6$ | $2.7 \times 10^6$ |

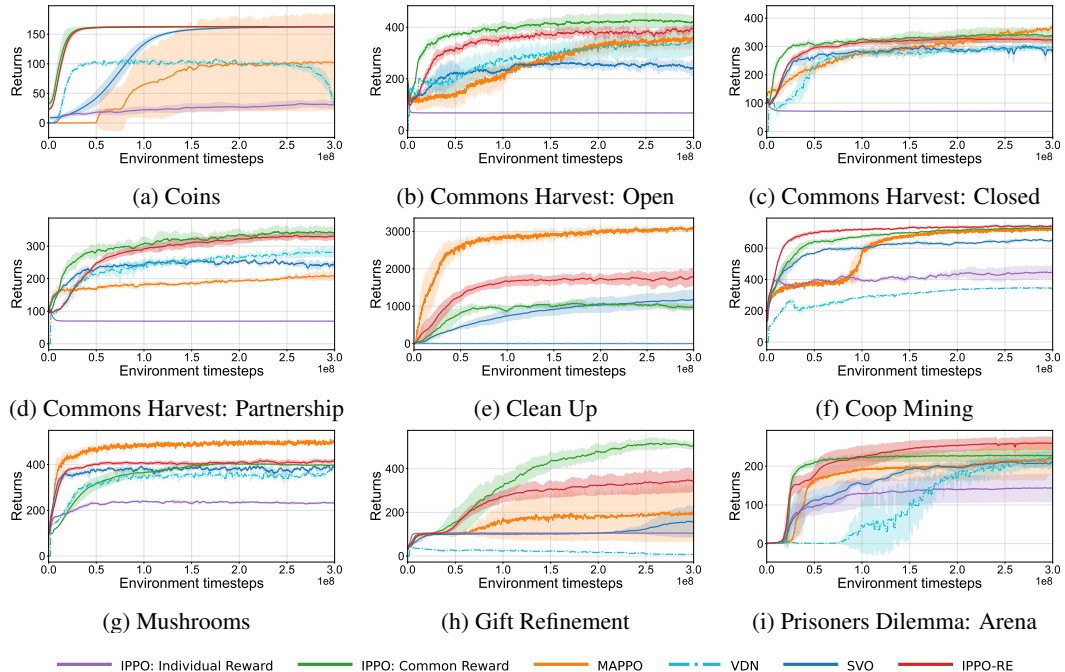

(a) Coins  (b) Commons Harvest: Open  (c) Commons Harvest: Closed

(d) Commons Harvest: Partnership  (e) Clean Up  (f) Coop Mining

(g) Mushrooms  (h) Gift Refinement  (i) Prisoners Dilemma: Arena

IPPO: Individual Reward    IPPO: Common Reward    MAPPO    VDN    SVO    IPPO-RE

Figure 2: Training curves for a range of SocialJax environments. IPPO (shared parameters) with Common Reward encourages collective interests, leading to higher overall returns, Individual Reward primarily drives selfish behavior, often resulting in lower returns. MAPPO is included as a centralized baseline. The SVO curve represents the case with a social angle $\theta = 90°$.

## 4.2 RESULTS

**Speed Benchmarks** Since both the environments and algorithms are vectorized using JAX, they are well suited for GPU acceleration. In Table 1, we test up to 4096 environments in parallel and observe a significant increase in the speed of the training pipeline with the addition of more parallel environments. We compare the performance of SocialJax and Melting Pot 2.0 (Agapiou et al., 2022) by measuring steps per second on the same hardware, which consists of an NVIDIA A100 GPU and 14 CPU cores. As shown in Table 1, we evaluate the speed of SocialJax across various environments and environment counts using random actions. We observe a significant speedup when running environments in parallel on a GPU compared to running a single environment. To match the parallel performance of SocialJax, the original environments would need to scale up across hundreds of CPUs. Regarding wall clock time, SocialJax is capable of completing 1e9 timesteps within 3 hours for relatively simple environments such as Coins, whereas training with Melting Pot 2.0 using Stable-Baselines3 and RLlib takes around 1,300 hours and 150 hours, respectively. Consequently, conducting social dilemma experiments with Melting Pot 2.0 is highly challenging without a large amount of CPU resources. Training Coins in SocialJax is about 50-400 times faster compared to Coins in Melting Pot 2.0. For relatively complex environments such as Clean Up, SocialJax is approximately 50-140 times faster than the original environments. There are more details about the wall clock time comparison in the Appendix C.2.

**IPPO Common reward and IPPO Individual reward** We compare the performance of common reward IPPO (IPPO-CR) and individual reward IPPO (IPPO-IR) across different environments. In the default *individual reward* setting, each agent receives a reward based on the joint-action, independent of the payoffs of other agents. This creates incentives for agents to over exploit common pool resources, because only they benefit from their actions, while the social costs are shared by all. In contrast, a *common reward*, or *team reward* means that all agents share a single reward signal, typically the total payoff of all agents in the environment. The common reward encourages cooperation, as the objective of each agent is to maximize the total reward, making individual agents more inclined to adopt collaborative behaviors. In Figure 2, a common reward typically performs better than individual rewards, as the agents attempt to maximize the total payoff.

Table 2: Returns Across Different Environments as a Function of $\theta$ under the SVO Algorithm.

| Environment | 0° (Individualistic) | 22.5° | 45° | 67.5° | 90° (Altruistic) |
|---|---|---|---|---|---|
| Coins | 11.81 | 60.38 | 160.43 | 162.25 | **162.46** |
| Harvest: Open | 79.71 | 68.96 | 68.28 | 238.40 | **254.54** |
| Harvest: Closed | 81.05 | 74.74 | 74.68 | 298.97 | **309.94** |
| Harvest: Partnership | 77.44 | 72.45 | 71.58 | 225.94 | **234.30** |
| Clean Up | 0.02 | 0.06 | 50.58 | 1060.25 | **1410.53** |
| Coop Mining | 210.26 | 207.96 | 415.99 | 630.32 | **647.61** |
| Mushrooms | 5.94 | 51.26 | 291.55 | 321.22 | **400.85** |
| Gift Refinement | 104.29 | 105.09 | 105.22 | 107.38 | **227.95** |
| Prisoner's Dilemma: Arena | 22.67 | 23.27 | 24.13 | 34.96 | **53.36** |

**MAPPO** As a centralized learning method, MAPPO has access to information from all agents, which leads to strong performance in Clean Up (Figure 2e), and Mushrooms (Figure 2g). However, MAPPO exhibits less stable training dynamics, as its centralized critic aggregates observations from all agents, increasing the learning difficulty of the network. In particular, it struggles with Commons Harvest: Partnership (Figure 2d) and Gift Refinement (Figure 2h), where it under performs IPPO-CR.

**VDN** outperforms the IPPO: Individual Reward baseline in several environments, including Coins, Commons Harvest variants, Mushrooms, and Prisoner's Dilemma: Arena (see Figures 2a to 2d, 2g and 2i), but its linear decomposition of the joint Q-value restricts its ability to model complex inter-agent interactions, leading to suboptimal performance in harder social dilemmas.

**SVO** provides a powerful framework for exploring agents' cooperative and competitive tendencies. We apply SVO across all SocialJax environments to investigate how the preference angle $\theta$ influences agents' returns. We first fix $\theta = 90°$ and sweep over different weight values $w$ to identify the optimal $w$. Then, holding the weight fixed, we sweep over $\theta$ to examine how agent behavior and returns vary with the angle. As summarized in Table 2, the collective return increases with larger values of $\theta$, as agents become more inclined to cooperate at higher angles. When $\theta = 0°$ (Individualistic orientation), agents achieve the lowest returns, whereas at $\theta = 90°$ (Altruistic orientation), they attain the highest returns.

**IPPO-RE** results are shown with the best-performing reward ratio in Figure 2. Reward exchange substantially improves over IPPO-CR in Clean Up (Figure 2e), with a modest improvement in Prisoner's Dilemma (Figure 2i), suggesting that performance in these environments benefits from retaining some individual incentives. However, in Gift Refinement (Figure 2h), reward exchange performs substantially worse than IPPO-CR, suggesting that individual incentives to free-ride are particularly strong in this environment.

## 4.3 Social Dilemma Environments Attributes

**Schelling Diagram Analysis** To validate the social dilemma properties of SocialJax, we depict Schelling diagrams for each environment. The cooperative policies are sampled from the agents that used a common reward, while the defector policies originate from agents trained with independent rewards. We evaluate the environments over 30 episodes and compute the average rewards for the cooperative and defecting agents.

Coins (Figure 3a), Harvest: Open (Figure 3b), Harvest: Closed (Figure 3c), Harvest: Partnership (Figure 3d) Gift Refinement (Figure 3h), Mushrooms (Figure 3g) and exhibit fear: namely that agents would prefer to defect if some of their co-players are defecting. An agent actually does better cooperating if all their co-players are cooperating. In Harvest: Open, we observe that it only takes a single defector agent to have a severely negative impact on the returns of other players. This is because defector agents typically harvest the last apple in a patch, permanently depleting it. The cooperator policies have learned to leave a greater amount of apples in each patch, which causes a higher regrowth rate, so they generally exhibit a greater tendency to abstain from harvesting in the presence of a defector. In Harvest: Closed, we find that more than one defector is necessary to inflict a severe negative impact on the other agents. This occurs because Harvest: Closed partitions resources into two distinct rooms. After a lone defector depletes all apples in one room, re-entering the other requires visually locating its entrance and traversing through it.

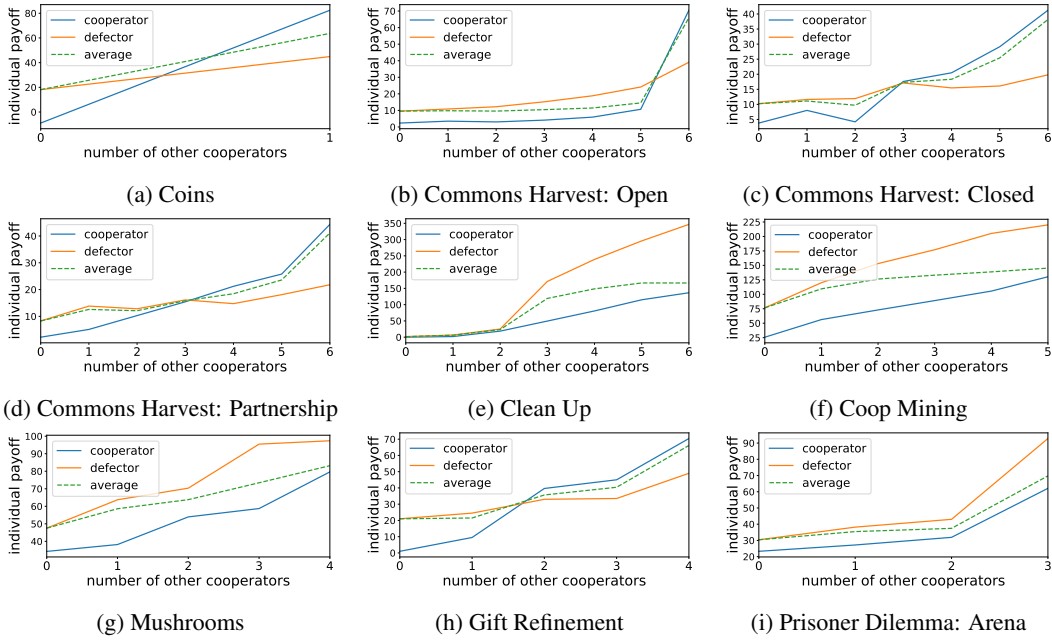

Figure 3: Schelling diagrams of the environments, visualizing the relationship between individual incentives and collective outcomes. The dotted line represents the overall average return of cooperators and defectors.

In Clean Up (Figure 3e), we instead observe greed, as there is an incentive to defect if a sufficient number of co-players are cooperating. This is because cooperators will clean the pollution, whereas the defector will not, but it requires a minimum of three cooperators to clean the river for apples to grow. Coop Mining (Figure 3f) and Prisoner's Dilemma: Arena (Figure 3i) also primarily exhibit greed. In these environments, if a sufficient number of agents cooperate, other agents can obtain a higher payoff by defecting. In Coop Mining, this is because cooperators tend to neglect the iron ore, allowing the defector to collect more iron ore. In Prisoner's Dilemma: Arena, when most agents collect cooperative resources, defectors can gain higher returns through interactions with cooperators.

**SVO Metrics Analysis** The proposed metrics quantify changes in agent behavior or environment states triggered by cooperative actions. A larger metric (Table 3) value reflects a higher level of cooperative behavior exhibited by the agents. In general, a higher degree of cooperative behavior among agents leads to larger collective returns. However, there are exceptions. In the three Harvest environments, the SVO algorithm encourages altruistic actions, which can reduce the agents' efficiency in collecting apples. Specifically, agents may intentionally avoid picking apples to allow others access to them. As a result, more apples remain on the map, leading to a decrease in harvest efficiency and ultimately a reduction in collective returns. In more extreme cases, agents may not collect any apples at all, causing the Harvest metric to remain at a high value. Under typical conditions where agents receive rewards based on apple collection, this metric performs well. As expected, IPPO-IR and SVO-$0°$, which encourage selfish behavior, produce significantly lower metric values than more cooperative approaches such as IPPO-CR, MAPPO and SVO-$90°$.

**Reward Exchange Analysis** While common reward fully aligns agent incentives, it weakens the link between an agent's actions and its utility, making it difficult for agents to identify which of their actions contributed to a reward (the credit assignment problem) and allowing agents to benefit from others' efforts without contributing themselves (the lazy agent problem). For each environment, we identify the minimum reward exchange needed to resolve the conflict between individual and collective incentives by selecting the smallest ratio achieving at least 95% of the best-performing ratio's return, to account for stochastic variation across seeds. Table 4 reports the results, with the selected ratio in bold.

For all environments, some reward exchange is necessary to achieve good outcomes. Coop Mining requires the least exchange (ratio 0.25). Harvest: Closed and Harvest: Partnership achieve good returns at ratio 0.75, but Harvest: Open requires a common reward, as restraining harvesting is harder

Table 3: Metrics for Assessing Cooperative and Competitive Tendencies Across Environments. IPPO-RE is implemented using the best-performing reward exchange ratio.

| Envs | IPPO-IR | IPPO-CR | MAPPO | VDN | IPPO-RE | SVO-90° |
|---|---|---|---|---|---|---|
| Coins | 100.83 | 164.07 | 126.23 | 15.06 | 164.06 | **164.56** |
| Harvest: Open | 0.91 | 23.68 | 18.45 | 21.41 | 28.57 | **37.67** |
| Harvest: Closed | 1.71 | 17.57 | 18.57 | 22.47 | **24.22** | 22.70 |
| Harvest: Partnership | 1.42 | 17.75 | 15.56 | **27.34** | 24.73 | 25.12 |
| Clean Up | 9.28 | 109.54 | **133.99** | 13.71 | 106.71 | 99.08 |
| Coop Mining | 298.21 | 521.56 | **571.25** | 285.16 | 204.19 | 483.24 |
| Mushrooms | 4.07 | 19.55 | **22.96** | 10.02 | 15.57 | 12.33 |
| Gift Refinement | 0.03 | **28.37** | 20.24 | 0.01 | 12.20 | 3.83 |
| Prisoner's Dilemma | 58.65 | 73.67 | **119.63** | 78.45 | 50.13 | 76.24 |

Table 4: IPPO-RE Mean return (± std) across environments under different reward exchange ratios. In bold, we highlight the smallest ratio that achieves at least 95% of the returns of the best ratio.

| Environment | Individual | Ratio 0.25 | Ratio 0.5 | Ratio 0.75 | Common Reward |
|---|---|---|---|---|---|
| Coins | $39 \pm 5$ | $93 \pm 1$ | $\mathbf{162 \pm 0}$ | $163 \pm 0$ | $164 \pm 0$ |
| Harvest: Open | $67 \pm 0$ | $70 \pm 1$ | $283 \pm 42$ | $393 \pm 21$ | $\mathbf{420 \pm 35}$ |
| Harvest: Closed | $71 \pm 1$ | $191 \pm 18$ | $311 \pm 30$ | $\mathbf{323 \pm 18}$ | $338 \pm 29$ |
| Harvest: Partnership | $69 \pm 0$ | $136 \pm 27$ | $292 \pm 17$ | $\mathbf{328 \pm 14}$ | $342 \pm 17$ |
| Clean Up | $0 \pm 0$ | $1260 \pm 87$ | $\mathbf{1780 \pm 340}$ | $1310 \pm 410$ | $970 \pm 90$ |
| Coop Mining | $446 \pm 49$ | $\mathbf{727 \pm 9}$ | $741 \pm 7$ | $729 \pm 17$ | $726 \pm 9$ |
| Mushrooms | $232 \pm 5$ | $342 \pm 65$ | $\mathbf{415 \pm 11}$ | $410 \pm 11$ | $398 \pm 1$ |
| Gift Refinement | $105 \pm 0$ | $184 \pm 27$ | $299 \pm 25$ | $342 \pm 65$ | $\mathbf{505 \pm 18}$ |
| Prisoner's Dilemma | $144 \pm 40$ | $233 \pm 48$ | $\mathbf{260 \pm 23}$ | $226 \pm 12$ | $227 \pm 17$ |

without spatial partitioning. Gift Refinement also requires a common reward and shows the largest performance drop from any reduction in reward sharing, suggesting that individual incentives to free-ride are particularly strong. Clean Up achieves its best returns at ratio 0.5—notably outperforming the common reward—consistent with the hypothesis that retaining some individual incentive alleviates credit assignment difficulties. The reward ratio at which cooperation emerges is related to the self-interest level of the game (Willis et al., 2024); we leave formal estimation in these environments to future work.

## 5 Conclusion

The use of hardware acceleration has revolutionized MARL research, enabling researchers to overcome computational limitations and accelerate the iteration of ideas. SocialJax is a library designed to leverage these advancements in the context of social dilemmas. It provides a collection of social dilemma environments and baseline algorithms implemented in JAX, offering user-friendly interfaces combined with the efficiency of hardware acceleration. By achieving remarkable speed-ups over traditional CPU-based implementations, SocialJax significantly reduces the experiment runtime. In addition, its unified codebase consolidates a diverse range of social dilemma environments into a standardized framework, facilitating consistent and efficient experimentation and greatly benefiting future research on social dilemmas.

## Acknowledgement

This work was supported by the Engineering and Physical Sciences Research Council [grant number EP/Y003187/1 and UKRI849].

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

## A  STATEMENTS OF ETHICS, REPRODUCIBILITY, AND LLM USAGE

This work complies with the ICLR Code of Ethics. No human or animal subjects were involved, and all datasets, including SocialJax environments and training outputs, were used according to relevant guidelines with no privacy violations. We took care to mitigate potential biases and ensured transparency, fairness, and integrity throughout. To support reproducibility, all code, datasets, and detailed experimental settings, including training steps, model configurations and hyperparameters, are publicly available in SocialJax repository. Large Language Models (LLMs) were used solely for linguistic refinement, such as improving clarity, grammar, and coherence; they did not contribute to research design, analysis, or conceptualization. The authors take full responsibility for all scientific content.

## B  ADDITIONAL DETAILS ON ENVIRONMENTS

### B.1  API AND EXAMPLES

Our environment interfaces (Figure 4) are inspired by those of mature MARL frameworks, adopting the similar API conventions as JaxMARL , which itself is inspired by PettingZoo and Gymnax, to provide an intuitive and user-friendly experience for researchers.

```python
import jax
import socialjax
from socialjax import make

num_agents = 7
env = make('clean_up', num_agents=num_agents)
rng = jax.random.PRNGKey(259)
rng, _rng = jax.random.split(rng)

for t in range(100):
    rng, *rngs = jax.random.split(rng, num_agents+1)
    actions = [jax.random.choice(
        rngs[a],
        a=env.action_space(0).n,
        p=jnp.array([0.1, 0.1, 0.1, 0.1, 0.1, 0.1, 0.1, 0.1, 0.1])
    ) for a in range(num_agents)]

    obs, state, reward, done, info = env.step_env(
        rng, old_state, [a for a in actions]
    )
```

Figure 4: An example of using the SocialJax API in the Clean Up environment.

Because we provide an open-source codebase, anyone can reproduce our main experimental results by configuring the environment following the repository's instructions and running the code. Figure 5 provides command-line examples for executing various algorithms.

Execution speed of SocialJax environments under random actions can also be evaluated by running the code shown in Figure 6. Additionally, we provide a Colab notebook in our repository for quick experimentation and environment speed testing.

### B.2  ENVIRONMENT SETTINGS AND PARAMETERS

The data format observed by agents in SocialJax environments differ from those in Melting Pot 2.0 (Agapiou et al., 2022). Melting Pot 2.0 provides observations to agents in the form of image pixels. Melting Pot 2.0 was originally designed to use $8 \times 8$ pixels per cell, resulting in observations of $88 \times 88$ pixels in most environments. In this setup, the agents have a partially observable window of $11 \times 11$ cells. In the Melting Pot Contest 2023 (Trivedi et al., 2024), all academic participants down-sampled

```
# Train IPPO on the Clean Up environment
python algorithms/IPPO/ippo_cnn_cleanup.py

# Train MAPPO on the Clean Up environment
python algorithms/MAPPO/mappo_cnn_cleanup.py

# Train SVO on the Clean Up environment
python algorithms/SVO/svo_cnn_cleanup.py
```

Figure 5: Command-line example for training different algorithms (IPPO and MAPPO) and generating training curves with SocialJax.

```
python speed_test/speed_test_random.py
```

Figure 6: Command-line example for testing SocialJax environment execution speed with random actions.

the cells to $1 \times 1$, also producing observations of $11 \times 11$. The agents in our environments have the same $11 \times 11$ field of view, but they are implemented at the grid level.

**Coins** Two players collect coins in a shared room, where each coin is assigned to one player's color. Each coin, upon appearing, has an equal 50% probability of being assigned to the first player's color or the second player's color. An agent receives a reward of 1 for collecting any coin, regardless of its color. If one player collects a coin assigned to the other player's color, the other player receives a reward of -2. Each empty tile on the map has a respawn probability of $p = 0.0005$ for each type of coin.

**Commons Harvest** The core concept of Commons Harvest was first defined by (Janssen et al., 2010). (Pérolat et al., 2017) introduced it in multi-agent scenarios. Melting Pot 2.0 (Agapiou et al., 2022) has made it a more comprehensive environment. There are several patches of apples in the room. The players can receive a reward of 1, when they collect one apple. The apples will regrow with probability that is determined by the number of neighborhood apples in radius 2, when the apples are collected by the players. The probability of apple regrowth decreases as the nearby apples diminish. Apples will not regrow, if there are no apples in the neighborhood. Formally, the regrowth probability is set to $0.025$ if there are at least three apples nearby, $0.005$ if there are exactly two, $0.001$ if there is exactly one, and $0$ when there are no apples in the neighborhood. Once all apples within a patch are collected, that patch ceases to respawn apples.

*Commons Harvest: Open*: This experiment is conducted in an open environment with no internal obstacles, and walls are placed only along the periphery of the map.

*Commons Harvest: Closed*: the players' roles are divided into two groups: one focuses on collecting apples, while the other is tasked with preventing over-harvesting by other players (Pérolat et al., 2017). There are two rooms in this setup, each containing multiple patches of apples and a single entry. These rooms can be defended by specific players to prevent others from accessing the apples by zapping them with with a beam, causing them to respawn.

*Commons Harvest: Partnership* In this environment, two players are required to cooperate in defending apple patches. The agents can be tested on whether they can learn to trust their partners to jointly defend their shared territory from intrusion and act sustainably when managing shared resources.

**Clean Up** The agent can earn +1 reward by collecting each apple in Clean Up environment. Apples grow in an orchard and their regrowth depends on the cleanliness of a nearby river. Pollution accumulates in the river at a constant rate and once pollution surpasses a certain threshold, the apple

growth rate drops to zero. Players have the option to perform a cleaning action that removes small amounts of pollution from the river.

*River Pollution Generation*: During the first 50 time steps of the game, the river remains uncontaminated. After 50 time steps, there is a 0.5 probability that one tile of the river will become polluted. *Apple Respawning*: In the environment, let $dirtCount$ represent the number of currently polluted river tiles, and $riverCount$ represent the total number of river tiles, including both polluted and unpolluted ones. Then, the proportion of polluted river tiles is given by $dirtFraction = \frac{dirtCount}{riverCount}$. The probability of apple regrowth $P$ is given by:

$$P = \alpha \cdot \min\left(\max\left(\frac{dirtFraction - \theta_d}{\theta_r - \theta_d}, 0\right), 1.0\right),$$

where:

- $\theta_d$ is the depletion threshold, the lower bound of the pollution range.
- $\theta_r$ is the restoration threshold, the upper bound of the pollution range.
- $\alpha$ is the maximum apple growth rate.

This formula calculates the probability $P$ of apple regrowth based on the current pollution level $f$, with the value clipped between 0 and 1 to ensure valid probabilities.

**Coop Mining**    In this environment, two types of ore spawn randomly in empty spaces. Players can extract the ore to get reward. Iron ore (gray) can be mined individually and provides a reward of +1 upon extraction. In contrast, gold ore (yellow) requires coordinated mining by two to four players within a 3-step window, granting a reward of +8 to each participant. When a player begins to mine gold ore, the state of the ore changes to "partially mined" to indicate readiness for another player to assist. Visually, this is represented by a brighter shade of yellow. If no other player cooperates or too many players attempt to mine simultaneously, the ore reverts to its original state, and no reward is granted.

We set the respawn rate of iron ore and gold ore to ensure that the ores regenerate at appropriate rates, promoting agents to make both selfish and cooperative choices, which in turn guarantees that the environment represents a social dilemma. The respawn rates are defined as: $P_{\text{iron}} = 0.0004$ and $P_{\text{gold}} = 0.00016$. These rates are chosen to balance resource regeneration and agent behavior, encouraging both competition and collaboration.

**Mushrooms**    There are four types of mushrooms spread across the map, each offering different rewards when consumed. Eating a red mushroom gives a reward of 1 to the player who consumed it, while eating a green mushroom gives a reward of 2, which is divided equally among all players. Consuming a blue mushroom grants a reward of 3, but the reward is shared among all players except the one who ate it. Additionally, eating an orange mushroom causes every agent to get a -0.2 reward.

Mushroom regrowth depends on the type of mushroom eaten by players. Red mushrooms regrow with a probability of 0.25 when any type of mushroom is eaten. Green mushrooms regrow with a probability of 0.4 when either a green or blue mushroom is eaten. Blue mushrooms regrow with a probability of 0.6 when a blue mushroom is eaten. Orange mushrooms always regrow with a probability of 1 when eaten. Each mushroom has a digestion time, and a agent who consumes a mushroom becomes frozen during the digestion process. Red mushrooms take 10 steps to digest, while green mushrooms take 15 steps to digest, and blue mushrooms take 20 steps to digest.

**Gift Refinement**    In this environment, tokens spawn on the map with probability $p = 0.0002$. When tokens are collected, they are stored in the agent's inventory. These stored tokens can be gifted to other agents, in which case the tokens are refined to a higher level and their quantity is tripled. Tokens exist in three levels, with only the lowest level spawning naturally in the environment. Each token can be refined at most twice to reach the highest level. Each agent can hold up to 15 tokens of each level. When agents execute the `consume` action, all tokens in the agent's inventory are converted into reward, with each token providing +1 reward regardless of refinement level.

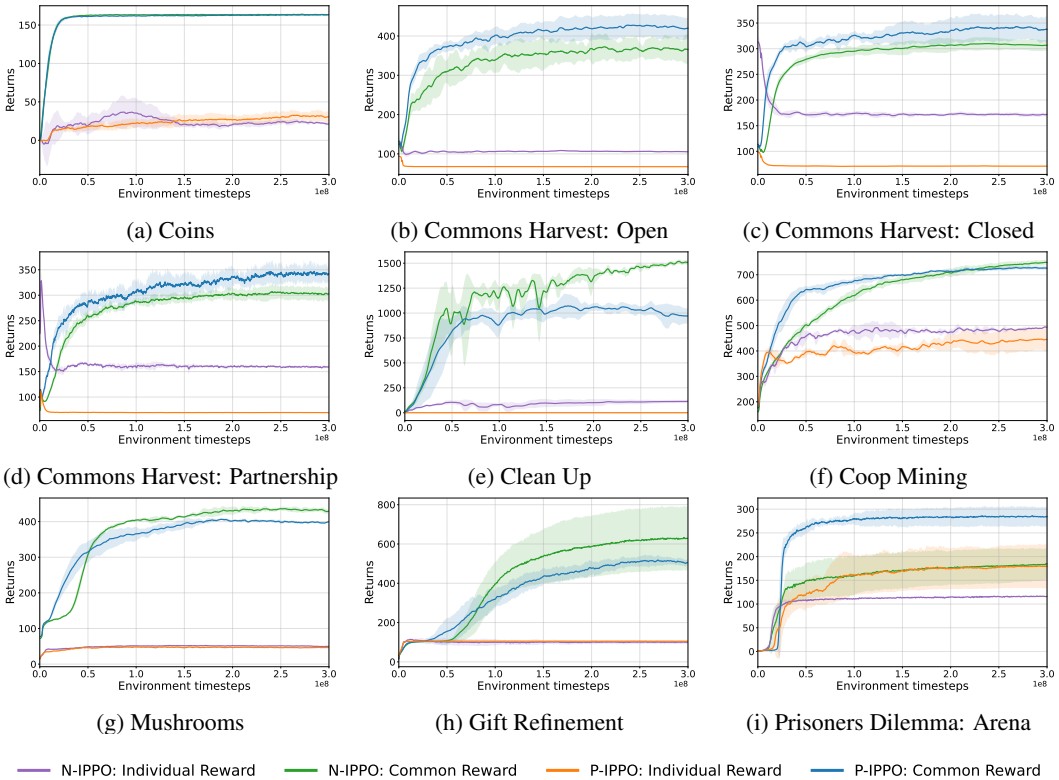

Figure 7: Training curves of IPPO Common Reward and IPPO Individual Reward. Common Reward encourages collective interests, leading to higher overall returns, while Individual Reward primarily drives selfish behavior, often resulting in lower returns.

**Prisoner's Dilemma: Arena**  Four individuals collect resources that represent 'defect' (red) or 'cooperate' (green) and compare inventories in an encounter, which is first introduced by (Vezhnevets et al., 2020). The inventory is represented by $\boldsymbol{\rho} = (\rho_1, \dots, \rho_K)$. In our case, $K = 2$ because we have two types of resources. When one agent zaps another, agents' inventories begin tracking anew, and each agent immediately receives its corresponding reward. Consequences of the inventory comparison are congruent with the classic Prisoner's Dilemma matrix game. This game exposes tension between reward for the group and reward for the individual. The matrix for the interaction is

$$A_{\text{row}} = A_{\text{col}}^T = \begin{bmatrix} 3 & -1 \\ 5 & 1 \end{bmatrix}.$$

Each agent has a mixed strategy weight, which is defined as

$$\mathbf{v} = (v_1, \dots, v_K), \quad v_i = \frac{\rho_i}{\sum_{j=1}^K \rho_j}.$$

When one agent (the "row player") zaps and another (the "column player") is targeted, their respective payoffs are given by bilinear forms over these mixed strategies:

$$r_{\text{row}} = \mathbf{v}_{\text{row}}^{\mathsf{T}} A_{\text{row}} \mathbf{v}_{\text{col}}, \quad r_{\text{col}} = \mathbf{v}_{\text{row}}^{\mathsf{T}} A_{\text{col}} \mathbf{v}_{\text{col}}.$$

After two agents complete an interaction, they respawn at designated locations within the environment and are prohibited from moving for a random duration of 10 to 100 time steps.

### B.3 DIVERSITY OF SOCIALJAX ENVIRONMENTS

Table 5 maps classical social dilemmas to the environments studied in our benchmark. Each paradigm captures a distinct cooperation–defection tradeoff, which is instantiated in specific multi-agent

environments. For instance, in the Prisoner's Dilemma, mutual cooperation strictly dominates mutual defection, yet unilateral defection remains individually tempting (e.g., Arena, Coins). In the Snowdrift/Chicken Game, cooperation yields benefits even if the partner defects, though mutual cooperation is preferable (e.g., Mushrooms). The Stag Hunt requires mutual cooperation to unlock high payoffs, while unilateral action results in low returns (e.g., Coop Mining, Gift Refinement). Finally, the Tragedy of the Commons illustrates the overuse of shared resources under selfish behavior, leading to system degradation (e.g., Commons Harvest variants, Clean Up).

Table 5: Mapping classical social dilemmas to SocialJax environments.

| Paradigm | Mapped Environments |
| --- | --- |
| Prisoner's Dilemma | Prisoner's Dilemma: Arena; Coins |
| Snowdrift / Chicken Game | Mushrooms |
| Stag Hunt | Coop Mining; Gift Refinement |
| Tragedy of the Commons | Commons Harvest (Open / Closed / Partnership); Clean Up |

### B.4 TASK DIFFICULTY CATEGORIZATION

we include a qualitative categorization Table 6 of the task environments used in our study along three axes: long-term impact of decisions, task multi-modality (i.e., whether agents need to solve multiple goals), and number of agents. These axes provide intuitive guidance on the sources of complexity across different social dilemmas. The final "Difficulty" column provides a coarse summary based on the combination of these factors.

Table 6: Environments and their characteristics.

| Environment | Long-Term Impacts | Multi-Task | Agents Number | Difficulty |
| --- | --- | --- | --- | --- |
| Clean Up | High | Yes | 7 | Very Hard |
| Gift Refinement | High | Yes | 5 | Very Hard |
| Harvest: Partnership | High | No | 7 | Hard |
| Harvest: Closed | High | No | 7 | Hard |
| Harvest: Open | High | No | 7 | Medium |
| Coop Mining | Medium | No | 6 | Medium |
| Prisoner's Dilemma | Medium | Yes | 5 | Medium |
| Mushrooms | High | No | 5 | Easy |
| Coins | Low | No | 2 | Easy |

### B.5 SCALABILITY OF SOCIALJAX

our framework allows users to freely adjust the number of agents via configuration files and environment settings, enabling flexible scaling experiments. We conducted additional tests to evaluate the environment speed with 20 agents under random action execution, excluding Coins and Prisoner's Dilemma due to their fixed design constraints, seeing Table 7. We observed only a modest slowdown in FPS compared to the performance reported in the main paper with original agent numbers. This is because our JAX-based environments rely heavily on parallel computation, where increasing the number of agents primarily affects GPU memory usage rather than significantly impacting computational speed.

## C ADDITIONAL RESULTS

### C.1 PARAMETER SHARING AND NON-PARAMETER SHARING IPPO

We compare the training time and learning performance of shared parameter IPPO (P-IPPO) versus non-shared parameter IPPO (N-IPPO). According to Figure 8, the training times of shared parameter

Table 7: Evaluate the environment speed with 20 agents under random action execution.

| Envs (20 Agents) | 1 JAX Env | 128 JAX Env | 1024 JAX Env | 4096 JAX Env |
|---|---|---|---|---|
| Harvest: Open | $1.2 \times 10^3$ | $1.2 \times 10^5$ | $5.1 \times 10^5$ | $7.8 \times 10^5$ |
| Harvest: Closed | $1.2 \times 10^3$ | $1.2 \times 10^5$ | $5.1 \times 10^5$ | $7.8 \times 10^5$ |
| Harvest: Partnership | $1.2 \times 10^3$ | $1.2 \times 10^5$ | $5.1 \times 10^5$ | $7.8 \times 10^5$ |
| Clean Up | $1.9 \times 10^3$ | $1.9 \times 10^5$ | $7.7 \times 10^5$ | $1.3 \times 10^6$ |
| Coop Mining | $1.9 \times 10^3$ | $2.0 \times 10^5$ | $9.8 \times 10^5$ | $1.5 \times 10^6$ |
| Mushrooms | $1.2 \times 10^3$ | $1.2 \times 10^5$ | $5.8 \times 10^5$ | $9.5 \times 10^5$ |
| Gift Refinement | $1.3 \times 10^3$ | $1.4 \times 10^5$ | $6.6 \times 10^5$ | $1.2 \times 10^6$ |

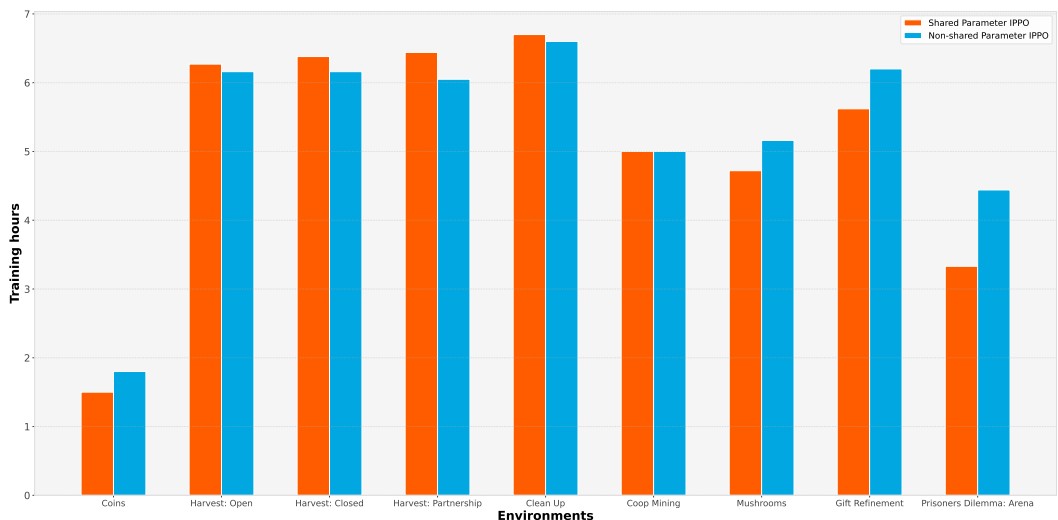

Figure 8: Training time to reach $10^9$ timesteps: Shared parameter IPPO versus Non-shared parameter IPPO.

IPPO and non-shared parameter IPPO are virtually indistinguishable, with the shared parameter variant completing slightly faster.

We also present the training curves Figure 7 for shared parameter and non-shared parameter IPPO, which demonstrate essentially identical convergence behavior.

## C.2 WALL CLOCK TIME SPEED COMPARISON

We compare the performance of our training pipeline with the training speed of the Stable-Baselines3 and RLlib implementations with TensorFlow, based on the publicly available code from Melting Pot 2.0.

Note that the experimental results in Melting Pot 2.0 (Agapiou et al., 2022) cannot be directly compared, as the code related to their reported results has not been released. The open-source versions of the algorithms in Melting Pot 2.0 are based on RLlib and Stable Baselines3. However, the versions of Actor-Critic Baseline (ACB) (Espeholt et al., 2018), V-MPO (Song et al., 2020), and OPRE (Vezhnevets et al., 2020) used in their reports have not been made available as open source. Additionally, since Melting Pot 2.0 is pixel-based environments while our SocialJax environments are grid-based, it is not feasible to align the environment and neural network settings directly. Therefore, we only compare wall-clock training times for scenarios that achieve convergence, evaluating the open-source frameworks Stable Baselines3 and RLlib alongside our own training pipeline.

As shown in Figure 9, the SocialJax training pipelines achieve roughly a 30–800× speedup over the open-source Melting Pot code, thereby substantially enhancing training efficiency. It should be noted

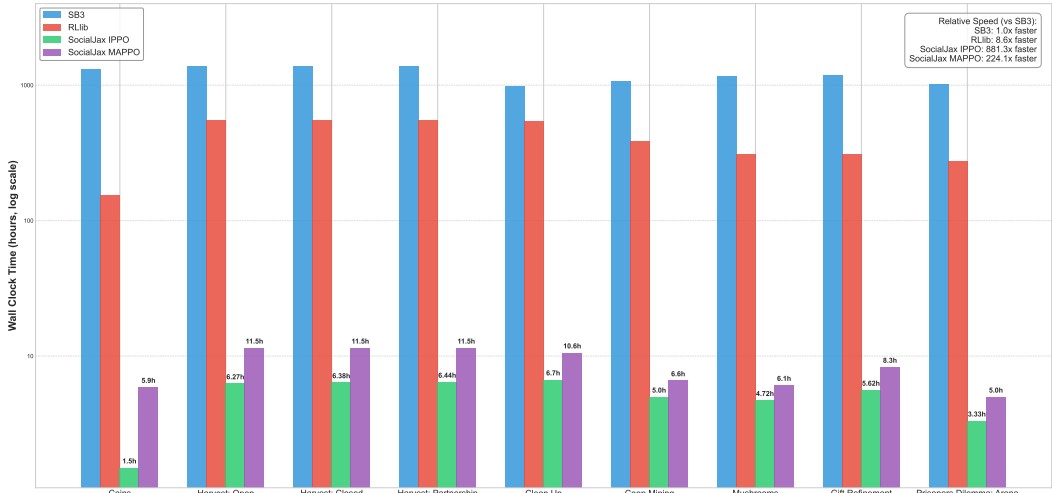

Figure 9: Training time comparison across different frameworks: Stable Baselines3, RLlib, IPPO, and MAPPO.

that we did not actually execute $10^9$ timesteps under the Stable Baselines3 and Rllib framework; rather, we estimated the time required to complete $10^9$ timesteps by extrapolating from the measured frames-per-second (FPS) performance.

## D HYPERPARAMETERS FOR TRAINING

In this section, we provide the hyperparameters and configuration details for different algorithms used in our environments. We first provide the hyperparameter settings for IPPO, along with configuration options for enabling or disabling parameter sharing, and for choosing between individual and common reward schemes. Table 8 shows the configuration of the IPPO algorithm. When shared_rewards is set to true, it corresponds to the common reward setting; when set to false, it corresponds to the individual reward setting. When PARAMETER_SHARING is true, it denotes shared parameter IPPO, and when false, non-shared parameter IPPO.

| Parameter | Value |
|---|---|
| Learning Rate | LR = 0.0005 |
| Number of Environments | NUM_ENVS = 256 |
| Number of Steps per Update | NUM_STEPS = 1000 |
| Total Timesteps | TOTAL_TIMESTEPS = $3 \times 10^8$ |
| Update Epochs | UPDATE_EPOCHS = 2 |
| Number of Minibatches | NUM_MINIBATCHES = 500 |
| Discount Factor | $\gamma = 0.99$ |
| GAE Lambda | GAE_LAMBDA = 0.95 |
| Clip Epsilon | CLIP_EPS = 0.2 |
| Entropy Coefficient | ENT_COEF = 0.01 |
| Value Function Coefficient | VF_COEF = 0.5 |
| Max Gradient Norm | MAX_GRAD_NORM = 0.5 |
| Activation Function | ACTIVATION = relu |
| Environment Name | ENV_NAME = clean_up |
| Reward Shaping Horizon | REW_SHAPING_HORIZON = $2.5 \times 10^6$ |
| Shaping Begin | SHAPING_BEGIN = $1 \times 10^6$ |
| Shared Rewards | shared_rewards = `False` (set `True` for common rewards) |
| CNN Enabled | cnn = `True` |
| JIT Compilation | jit = `True` |
| Anneal Learning Rate | ANNEAL_LR = `True` |
| Seed | SEED = 30 |
| Number of Seeds | NUM_SEEDS = 1 |
| Hyperparameter Tuning Enabled | TUNE = `True` |
| GIF Number of Frames | GIF_NUM_FRAMES = 250 |
| Parameter Sharing | PARAMETER_SHARING = `True` |
| Number of Agents | num_agents = 7 |
| Number of Inner Steps | num_inner_steps = 1000 |

Table 8: Hyperparameter settings and environment configuration for IPPO in the Clean Up environment.

| Description | Value |
|---|---|
| Learning Rate | LR = 0.001 |
| Number of Environments | NUM_ENVS = 4 |
| Number of Steps per Update | NUM_STEPS = 100 |
| Total Timesteps | TOTAL_TIMESTEPS = $3 \times 10^8$ |
| Update Epochs | UPDATE_EPOCHS = 2 |
| Number of Minibatches | NUM_MINIBATCHES = 4 |
| Discount Factor | $\gamma = 0.99$ |
| GAE Lambda | 0.95 |
| Clip Epsilon | CLIP_EPS = 0.2 |
| Entropy Coefficient | ENT_COEF = 0.01 |
| Value Function Coefficient | VF_COEF = 0.5 |
| Max Gradient Norm | MAX_GRAD_NORM = 0.5 |
| Scale Clip Epsilon | True |
| Activation Function | relu |
| Environment Name | "clean_up" |
| Reward Shaping Horizon | $2.5 \times 10^6$ |
| Number of Agents | 7 |
| Number of Inner Steps | 1000 |
| Shared Rewards | True — set to False for individual rewards |
| CNN Enabled | True |
| JIT Compilation | True |
| Anneal Learning Rate | True |

Table 9: Hyperparameter settings and environment configuration for MAPPO in the Clean Up environment.

Table 9 presents the hyperparameters for the MAPPO algorithm. For the SVO algorithm, our hyperparameters are shown in Table 10. Note that during SVO training, we first perform a sweep over the parameter $w$ to find its optimal value, and then conduct a sweep over the angle $\theta$. Therefore, the training script needs to be configured to sequentially perform a sweep over the parameter $w$ (Table 10) followed by a sweep over the angle $\theta$ (Table 11). Further details on environment configurations and hyperparameter settings can be found in our open-source implementation.

```
sweep_config = {
    "name": "cleanup_w",
    "method": "grid",
    "metric": {
      "name": "returned_episode_original_returns",
      "goal": "maximize",
    },
    "parameters": {
      "ENV_KWARGS.svo_w": {"values": [0.0, 0.2, 0.4, 0.6, 0.8, 1.0]},
    },
}
```

Figure 10: Sweep configuration for SVO weight $w$ in the Clean Up environment.

```
sweep_config = {
    "name": "cleanup_angle",
    "method": "grid",
    "metric": {
        "name": "returned_episode_original_returns",
        "goal": "maximize",
    },
    "parameters": {
        "ENV_KWARGS.svo_ideal_angle_degrees": {
            "values": [0, 22.5, 45, 67.5, 90]
        },
    },
}
```

Figure 11: Sweep configuration for SVO angle $\theta$ in the Clean Up environment.

| Parameter | Value |
|---|---|
| Learning Rate | LR=0.0005 |
| Number of Environments | NUM_ENVS=256 |
| Steps per Update | NUM_STEPS=1000 |
| Total Timesteps | TOTAL_TIMESTEPS=$3 \times 10^8$ |
| Update Epochs | UPDATE_EPOCHS=2 |
| Minibatches | NUM_MINIBATCHES=500 |
| Discount Factor | GAMMA=0.99 |
| GAE Lambda | GAE_LAMBDA=0.95 |
| Clip Epsilon | CLIP_EPS=0.2 |
| Entropy Coefficient | ENT_COEF=0.01 |
| Value Function Coefficient | VF_COEF=0.5 |
| Max Gradient Norm | MAX_GRAD_NORM=0.5 |
| Activation Function | ACTIVATION=relu |
| Environment Name | ENV_NAME=clean_up |
| Reward Shaping Horizon | REW_SHAPING_HORIZON=$2.5 \times 10^6$ |
| Number of Agents | ENV_KWARGS.num_agents=7 |
| Inner Steps | ENV_KWARGS.num_inner_steps=1000 |
| Shared Rewards | ENV_KWARGS.shared_rewards=False |
| CNN Enabled | ENV_KWARGS.cnn=True |
| JIT Compilation | ENV_KWARGS.jit=True |
| SVO Enabled | ENV_KWARGS.svo=True |
| SVO Target Agents | ENV_KWARGS.svo_target_agents=[0,1,2,3,4,5,6] |
| SVO Weight | ENV_KWARGS.svo_w=0.5 |
| SVO Ideal Angle (degrees) | ENV_KWARGS.svo_ideal_angle_degrees=90 |
| Anneal Learning Rate | ANNEAL_LR=False |
| Random Seed | SEED=30 |
| Number of Seeds | NUM_SEEDS=1 |
| Hyperparameter Tuning | TUNE=False |
| Reward Type | REWARD=individual |
| GIF Frame Count | GIF_NUM_FRAMES=250 |

Table 10: Hyperparameter settings and environment configuration for SVO training in the Clean Up environment.

