# OpenReview forum: "SocialJax: An Evaluation Suite for Multi-agent Reinforcement Learning in Sequential Social Dilemmas"
_ICLR.cc/2026/Conference — ICLR 2026 Poster_

### Official Review · Reviewer_m7Ji · 2025-10-28

**Soundness:** 3
**Presentation:** 3
**Contribution:** 3
**Rating:** 4
**Confidence:** 2

**Summary:**

This paper implements a series of sequential social dilemma environments in JAX, each exhibiting mixed incentive dynamics and significantly improving the speed of environment simulation. A range of multi-agent reinforcement learning (MARL) algorithms are implemented, and training these algorithms in SocialJAX is at least 50 times faster than in MeltingPot2.0. The performance of various environments and algorithms is benchmarked in terms of both simulation throughput and training speed.

**Strengths:**

The main contribution of this article is the implementation of a series of reinforcement learning algorithms using JAX, along with experiments and a comprehensive analysis.
If these works can be open-sourced, they would have practical significance for scenarios in the industry where there is a high demand for intelligent agents.

**Weaknesses:**

1. The work presented in this paper involves implementing multi-agent algorithms in the field of sequential social dilemma environments using the JAX library, which has accelerated the execution speed of the algorithms. The focus seems to be more on engineering aspects rather than innovation. The paper does not claim to have made any specific innovative engineering contributions that would have sped up the algorithms during implementation. It appears that the performance improvements are largely due to the advanced features of the JAX library.

2. This work is not declared open-source, and the links provided in the abstract cannot be accessed. This might be due to time constraints or the intention to add more information during the rebuttal phase, but it is unfair to other works. Not being open-source limits the contribution to the community, and the inability to view the links makes it impossible to evaluate the workload.

**Questions:**

this paper may address its issues of innovation rather than providing more engineering descriptions. If it is an engineering contribution, more detailed information should be provided during the experimental phase.

---

> ### Author Response · Authors · 2025-11-22
>
> We appreciate the reviewer’s comments and would like to provide additional clarification regarding both our contributions and the accessibility of the submitted resources.
>
>
> **Weakness 1: The focus seems to be more on engineering aspects rather than innovation.**
>
> While JAX was designed to accelerate neural network computation, it was not intended for building RL environments. Implementing sequential social dilemma environments as compilable JAX functions is technically nontrivial and fundamentally different from conventional CPU-based environments. Designing such environments in JAX is therefore a challenge rather than a straightforward application of the library.
>
> Our contribution lies in designing and implementing a practical suite of social dilemma environments and corresponding algorithms. SocialJax provides a robust and scalable platform for studying social dilemmas, consisting of nine diverse environments and benchmark algorithms. We employ Schelling diagrams and analysis to verify and characterize the inherent social dilemma properties. We also provide an SVO-based analysis illustrating how different angles induce shifts between selfish and prosocial behavior, offering useful reference points for interpreting the behaviors of different agents in our environment.
>
> We hope that SocialJax will serve as a valuable resource for the community, supporting further research and the development of new algorithms.
>
> **Weakness 2: Link Issure**
>
> Our code is open-source and released under the Apache License 2.0.
>
> We have tested the anonymous GitHub link provided in our abstract under different network environments and browsers, and we were all able to access it successfully. Could you please check the line again? There is also a possibility that recent cloud flare network issues are causing instability when accessing anonymous GitHub. You may try again at a different time, as the service often recovers automatically. If you are still experiencing issues, could you please provide us with more detailed information so that we can assist you further?
>
> **Question 1: Engineering Contribution Description**
>
> *Why Environments Based on JAX Are Hard to Implement*
>
> While JAX provides autograd, JIT compilation, and composable functional transformations, implementing RL environments directly in JAX is substantially more complex than in standard CPU-based frameworks. When coding and implementing environments, it is often necessary to use conditionals, loops, and dynamic data structures (such as dictionaries or lists). However, these programming patterns cannot be directly used in JAX, because GPUs excel at parallel array operations but struggle with dynamic control flow and data structures.
>
> *How JAX Enables Environments Fully Run on GPUs*
>
> In JAX, an environment can be implemented as a collection of pure functions, with states and transitions represented using JAX arrays. Because JAX follows a functional programming paradigm, all environment logic (such as `step()` and `reset()`) can be compiled by XLA via `jax.jit` as long as it is expressed through traceable array operations. Control flow can be handled using primitives such as `jax.lax.scan` and `jax.lax.cond`, allowing loops (`jax.lax.scan`) and branches (`jax.lax.cond`) to be included in the compiled computation graph. Therefore, as long as the environment avoids Python dynamic data structures (e.g., lists or dictionaries) and runtime side effects, the entire environment simulation(including multi-step rollouts) can run fully on the GPU.
>
>
> *SocialJax Environment Design and Modeling*
>
> The environments are designed with a deep understanding of social dilemma dynamics and are implemented in full accordance with JAX’s programming constraints, where control-flow constructs such as `for` and `if` cannot be used. For example, in Coop Mining, the environment tracks contributors using an `ore_miners` array (shape = (R, C, max_miners)), manages the time window through a `partial_ore_countdown`, and uses `jax.lax.scan` to process mining actions sequentially and avoid concurrency conflicts, which allows complex multi-step cooperation to be expressed entirely in pure functional JAX.
>
> *API Design and SocialJax Usability*
>
> In addition, our environment interface follows the conventions of `Gymnax` and `JaxMARL`, making it straightforward for researchers to use the environments, adjust parameters, or even create new ones. We believe that the efficiency of SocialJax can significantly support researchers in studying social dilemmas by reducing computational overhead and enabling faster experimentation.

---

> ### Author Response · Authors · 2025-11-28
>
> Dear Reviewers,
>
> May we ask if our response addressed your concerns, and if you are able to access our link? We are happy to answer any further questions and assist in resolving any issues with the link.
>
> Best Regards,
>
> Authors

---

### Official Review · Reviewer_CDVN · 2025-10-29

**Soundness:** 1
**Presentation:** 2
**Contribution:** 1
**Rating:** 2
**Confidence:** 5

**Summary:**

This paper presents SocialJax, a new suite of sequential social dilemma environments and algorithms built with JAX to advance research in multi-agent reinforcement learning (MARL).  In the second part of the paper, the authors validate the performance of baseline algorithms and use Schelling diagrams to study social dilemma outcomes - this second part appears rather disjointed from the first part.

**Strengths:**

+ Standardised benchmarks for studying social dilemmas using MARL agents are useful for the community. This paper goes in that direction.

**Weaknesses:**

- The discussion about CPU-based parallelism is difficult to understand. In fact, it seems that the problem is not at the level of the environment for GPU parallelism, but at the level of the agent implementation. It is possible to run multiple agents in a distributed fashion as well.

- In the introduction, the authors also claim that the social dilemmas are highly challenging environments. This appears to be a rather stretched claim. There are several environments that are actually much more complex than the simple dilemmas supported by SocialJAX.

- The authors appear to mix the problem of model implementation and environment when they discuss MAPPO, etc. (see paragraph entitled “Implementation of Algorithms in JAX” in the Introduction). In fact, you could release a very fast library implementing MAPPO, PPO, etc. instead.

- The authors say that the existing social dilemmas are restricted by CPU performance. This is not true. In fact, the models themselves can currently be run using GPUs in theory, but there is no inherent speed-up for them given their nature.  It seems that there is a mixing between agent implementation and interaction between agents in an environment. You can use standard PyTorch and use the GPU for implementing the agents. JAX allows you to do some clever synchronization in distributed training, not sure if it is useful/essential for the models that are described in this paper.

- In any case, the authors do not provide a discussion of the actual implementation details that lead to a speedup with respect to a “CPU implementation”. Details about the implementation, parallelization, and synchronization are missing.

- The structure of the paper is unconvincing in itself. There is no reason to introduce social dilemmas (and the mathematical details) in Section 3. The paper is supposed to be about a simulator, not about the models that are well known and widely used by the community. The set of games implemented are also quite standard (perhaps that section could be moved to an Appendix).

- In Table 1, the authors should have reported information regarding training, not random actions. The authors should have reported results for an increased number of CPUs.

- It is unclear if the authors used the same GPU cores in the simulations as Agapiou et al. Also, these do not appear as meaningful comparisons given the fact that these are different training libraries (for example, it is not due to the implementation of Melting Pot, etc.).

- The discussion about the IPPO common reward is not relevant to this paper. It is a generic result orthogonal to the problem of the design of a MARL simulator.

- The results in Table 2 and 3 and Figure 3 appear unrelated to the goal of this work. These are unrelated to the evaluation of the simulator in itself. The analysis of the metric is also quite disjointed from the core goal of the paper.

- In the Appendix, the wall clock comparison is not meaningful since they are based on different implementations (some of them are also pixel-based).

Minor weaknesses:

- There are missing citations for certain games in Section 4.2.
- Arena is in bold in Section 4.2.

**Questions:**

- Can you provide some technical description about the design choices exploiting JAX characteristics in order to support GPU parallelism (also in comparison to the characteristics of the “standard” environments discussed in your paper)?
- Did you the same GPU cores as Agapiou et al? Are the performance figures taken from that paper?

---

> ### Author Response · Authors · 2025-11-22
>
> We appreciate the reviewers’ thoughtful comments. We find that most concerns stem from misunderstandings or incomplete information, which we clarify below.
>
> **Clarification and Contributions**
>
> We did not use MeltingPot’s environments or algorithms. We also did not use MeltingPot environments to only run JAX-based agents.
>
> Based on MeltingPot’s core design principles, we reimplemented a new environment suite (SocialJax) entirely in JAX. By "reimplement", we mean we built an entirely new environments suite rather than adapting CPU-based MeltingPot environments. ***Our implementation is rewritten from scratch in JAX and does not reuse nor denpend on MeltingPot code.***
>
> Our contribution is reimplementing both environments and agent algorithms in JAX, enabling fully end-to-end GPU-based training. We verify sequencial social dilemma (SSD) properties using Schelling diagrams and provide benchmarks across algorithms.
>
> MeltingPot was designed for a different hardware ecosystem, assuming CPUs were inexpensive and GPUs/TPUs costly. DeepMind (Agapiou et al.) relied on over 1,000 CPUs to train MeltingPot. This setup is no longer available even in large companies, making direct comparisons difficult.
>
> **Weakness 1: "...CPU-based Parallelism..."**
>
> SocialJax leverages JAX to implement both environment and algorithms on the GPU, avoiding CPU–GPU communication overhead. A single GPU can run thousands of environment instances in parallel (Table 1), whereas MeltingPot requires hundreds of CPU cores. Traditional RL pipelines can run distributed agents but environments remain CPU-bound. GPU-based parallelism is thus central to our contribution.
>
> **Weakness 2: "...more complex...social dilemmas"**
>
> Complexity arises not from state or action space, but from agents pursuing individual payoffs that may reduce group welfare. The challenge is to encourage cooperation via mechanisms or learning algorithms but not the state or action space.
>
> **Weakness 3: "...problem of Model implementation..."**
>
> SocialJax environments and algorithms are designed to work together on the GPU. Only when both the environments and algorithms all JAX—based and run on GPUs can make SocialJax fully achieve its acceleration benefits. Algorithms cannot run directly on CPU-based environments without losing acceleration benefits.
>
> **Weakness 4: "...restricted by CPU performance..."**
>
> Existing SSD are limited by CPU-based environment execution. Even if agent models run on GPUs, CPU-bound environments dominate runtime, with additional GPU–CPU communication overhead. Standard PyTorch can only accelerate the agents but cannot do anything to CPU-based environments.
>
> **Weakness 5: "...implementation details..."**
>
> JAX offers autograd, JIT compilation, and composable functional transformations, allowing user-defined environments to run efficiently on GPUs. By avoiding Python dynamic data structures (like `list`, `dict`) and using `jax.lax.scan`/`jax.lax.cond` (loops and branches), environment logic can be fully GPU-compiled.
>
> **Weakness 6: "...introduce social dilemmas..."**
>
> We rebuilt environments in JAX following the main MeltingPot principles, so SocialJax cannot be exactly identical to Meltingpot. Therefore, we need to formally define SSDs, and use Schelling diagrams to confirm that our environments exhibit the required properties.
>
> **Weakness 7: "Table 1...not random actions"**
>
> Random actions demonstrate GPU-based parallelism. Direct training comparisons with MeltingPot are unfair due to different agent models (like size difference of the networks), because MeltingPot is pixcel-based. Appendix C.2 provides wall-clock comparisons corresponding to actual training. Besides, benchmarking MeltingPot across CPU counts is not our goal.
>
> **Weakness 8: "GPU cores"**
>
> All efficiency results are obtained with a NVIDIA A100 GPU and 14 CPU cores, as stated in Section 4.2 RESULTS of the paper.
>
> **Weakness 9: "IPPO common"**
>
> IPPO common evaluates full cooperation; IPPO individual evaluates full competition. Other algorithms balance cooperation and competition, covering fully cooperative, mixed, and fully competitive settings.
>
> **Weakness 10: "The results in Table 2 and 3 and Figure 3"**
>
> Because the SocialJax environments only shared core design principles with MeltingPot and rebuilt using JAX, which makes SocialJax are not identical to MeltingPot. So we need to prove our environments are SSDs. Tables 2, 3, and Figure 3 empirically demonstrate that our environments exhibit SSD properties under different algorithms and SVO settings. Schelling diagrams verify these SSD characteristics.
>
> **Weakness 11: "Wall Clock Comparison"**
>
> Wall clock comparisons are imperfect but illustrate that JAX-based design reduces training time by an order of magnitude on the same hardware, making SSD research more feasible for researchers who want to validate their algorithms
>
> **Minor weaknesses**
> Thank you so much for pointing out the minor weaknesses, we will fix them in next version.

---

> > ### Author Response · Authors · 2025-11-22
> > **Respond to the Questions**
> >
> > **Q1: Technical Description**
> >
> > We do not use CPU-based programming methods, such as using `for`, `if`, etc., to implement the logic. We must use JAX-based programming, which allows our environments and algorithms to run in parallel and be compiled to the GPU.
> > JAX uses function primitives such as `jax.lax.fori_loop`, `jax.lax.while_loop`, `jax.lax.scan` and `jax.lax.cond` to transform traditional imperative loops and branch conditions into a traceable functional programming style. These primitives are compatible with JIT compilation and can be converted by the XLA compiler into efficient GPU computation graphs. Our environments adopt the programming approach described above, enabling them to implement the full environment dynamics and interaction logic while remaining compatible with JAX’s requirements for GPU. Therefore, our environments run almost entirely on the GPU.
> >
> > **Q2: GPU cores and Performance Figures**
> >
> > Results were obtained on a A100 GPU and 14 CPU cores. We did not use Agapiou et al.’s hardware, which is no longer available. All performance figures come from our SocialJax implementation, which is newly built using MeltingPot design principles.
> >
> > We will add more details on the difficulty and implementation in the revision to explain our work. We are happy to answer any further questions. Thank you so much.

---

> ### Author Response · Authors · 2025-11-28
>
> Dear Reviewers,
>
> Could you please let us know whether we have adequately addressed your concerns? We would love to engage and answer any further questions.
>
> Best Regards,
> Authors

---

### Official Review · Reviewer_A4zv · 2025-10-30

**Soundness:** 3
**Presentation:** 4
**Contribution:** 3
**Rating:** 8
**Confidence:** 4

**Summary:**

This paper introduces SocialJax, a JAX-based suite of nine sequential social-dilemma environments plus baseline MARL implementations (IPPO under individual/common reward, MAPPO, SVO, PPO-RE). The environments are grid-based abstractions inspired by Melting Pot (e.g., Coins, Commons Harvest variants, Clean Up, Coop Mining, Mushrooms, Gift Refinement, Prisoner’s Dilemma: Arena), with a fixed 11×11 partially observable window. The authors emphasize throughput and reproducibility: by having thousands of vectorized environments on GPU this paper shows significant speedups by using SocialJax that allow reproducibility with significantly less computational resources. They validate “social dilemma” structure via Schelling diagrams built from policy classes: “cooperators” sampled from common-reward training; “defectors” from individual-reward training, and propose simple, environment-specific cooperation metrics beyond return.

**Strengths:**

The paper is well written and introduces a valuable benchmark for the MARL community. A major bottleneck of MARL research is associated with the large costs of training often available only to big industrial labs. In such a way, SocialJax is a step towards the democratization of MARL research by allowing fast training of important benchmarks with significantly lower computational constraints. I believe the main strength of this paper lies on the opportunity it might enable to push the frontiers of MARL research by making it more affordable and thus it should be accepted to the conference.

**Weaknesses:**

A core weakness is that SocialJax reduces the observation resolution and modality relative to the original environments, shifting from pixel observations (e.g., Melting Pot’s ~88×88 image crops for an 11×11 field of view) to discrete 11×11 grid values rendered at the grid level. This removes the perception-learning burden that makes the originals attractive for testing MARL scalability and representation learning, and it changes coordination difficulty and partial observability in ways that can inflate apparent sample efficiency. In the authors’ own description, SocialJax agents “observe a grid matrix” with a fixed 11×11 view, whereas Melting Pot provides pixel images; even Melting Pot contest entrants who downsampled still operated in a pixel pipeline, not a hand-engineered grid state. Consequently, speedups and learning curves are not apples-to-apples comparisons.

**Minor errors/typos:**

**Line 271:** The authors state “we evaluate cooperation by counting how many coins each agent collects that match their assigned color”. This is not a valid metric of cooperation as the agents may be learning to collect all the coins. A good metric of cooperation in this environment is “total coins of their assigned color”/ ”total coins collected” i.e. which fraction of the coins collected are of their color.

**Line 192, 768:** Duplicated word “with with a beam”.

**Questions:**

How well do the different algorithm implementation performances align with those in the original environments (e.g. IPPO in meltinpot’s common harvest open vs socialjax's common harvest open)?

Some results are unintuitive to me. In the Schelling diagrams, why is the reward of defectors lower than that of cooperators in Gift refinement as the number of other cooperators increases? What are the policies that the cooperators are learning?

---

> ### Author Response · Authors · 2025-11-22
>
> We appreciate the reviewer’s thoughtful analysis regarding the observation space differences between MeltingPot and SocialJax. We fully acknowledge that shifting from pixel-based observations to grid-based representations introduces differences in state dimensionality and perception complexity. Below, we clarify our design motivations, technical constraints, and explaination of Schelling diagram.
>
> **Weakness 1: observation resolution from pixel observations to discrete 11×11 grid values rendered at the grid level**
>
> > *A core weakness is that SocialJax reduces the observation resolution and modality relative to the original environments, shifting from pixel observations (e.g., Melting Pot’s ~88×88 image crops for an 11×11 field of view) to discrete 11×11 grid values rendered at the grid level. This removes the perception-learning burden that makes the originals attractive for testing MARL scalability and representation learning, and it changes coordination difficulty and partial observability in ways that can inflate apparent sample efficiency. In the authors’ own description, SocialJax agents “observe a grid matrix” with a fixed 11×11 view, whereas Melting Pot provides pixel images; even Melting Pot contest entrants who downsampled still operated in a pixel pipeline, not a hand-engineered grid state. Consequently, speedups and learning curves are not apples-to-apples comparisons.*
>
> We agree that the pixel-based MeltingPot and the grid-based SocialJax differ. The grid-based design indeed reduces the input dimensionality for neural networks makes the environments to be easier. However, we emphasize that the challenge represented by social dilemmas comes from the mixed-motive reward structure, rather than whether neural networks can interpret pixel-based maps. Our goal is to leverage MeltingPot’s core design principles to build efficient and user-friendly environments and algorithms.
>
> Due to differences in code architecture (in JAX we cannot use standard coding constructs like `for`, `if`, or dynamic variables like `list`), we cannot guarantee that SocialJax is identical to MeltingPot. Nonetheless, the core design principles are preserved and we proved that our environments are social dilemmas.
>
> While wall clock comparisons are imperfect, but we want to emphasize that SocialJax enables an exponential efficiency improvement for evaluating algorithms addressing social dilemmas, which can help the community test and validate methods much faster.
>
> **Minor errors/typos**
>
> Thank you very much. We agree with you and will address these issues accordingly.
>
> **Question 1: different algorithm implementation performances align with those in the original environments**
>
> > *How well do the different algorithm implementation performances align with those in the original environments (e.g. IPPO in meltinpot’s common harvest open vs socialjax's common harvest open)?*
>
> Since MeltingPot’s A3C-type algorithms are not open-sourced, we cannot reproduce the official reported results of MeltingPot. The codebase only provides a simple PPO implementation without official baseline results, so we need to build IPPO baseline for Meltingpot. When running PPO from MeltingPot in RLlib, we measured an FPS of 500–600 (A100 GPU, 14 CPU cores), which means reaching the same number of timesteps as our experiments would take 6–7 days for a single environment. Furthermore, our environments are not identical to MeltingPot’s, but we have verified that they exhibit social dilemma properties. We believe this provides a solid foundation for researchers to conduct algorithms using our environments.
>
> While we understand the reviewers’ concern, we cannot run all baselines on MeltingPot. This limitation underscores the challenges of MeltingPot in terms of reproducibility and the lack of open-source benchmarks, which motivated us to develop SocialJax.
>
> **Question 2: Schelling diagram of Gift refinement**
>
> > *Some results are unintuitive to me. In the Schelling diagrams, why is the reward of defectors lower than that of cooperators in Gift refinement as the number of other cooperators increases? What are the policies that the cooperators are learning?*
>
> In the Gifts environment, cooperators learn to act together. Once a cooperator collects a token, cooperators stop moving and give tokens to each other to maximize the team reward. Defectors, on the other hand, reject the giving action and tend to act individually to collect tokens on the map. As a result, cooperators rarely have the chance to approach defectors and give them tokens. In this scenario, as the number of cooperators increases, the reward of cooperators surpasses that of defectors.

---

> > ### Comment · Reviewer_A4zv · 2025-11-25
> >
> > I thank the authors for answering my concerns and questions and in view of this will maintain my current score.

---

> > > ### Author Response · Authors · 2025-11-28
> > >
> > > We are very grateful for your help, encouragement, and support.

---

### Official Review · Reviewer_RUpe · 2025-10-31

**Soundness:** 3
**Presentation:** 3
**Contribution:** 2
**Rating:** 6
**Confidence:** 2

**Summary:**

The authors described an evaluation suite for multi-agent RL in social dilemmas.
This suite consisted of 9 RL environments exploring cooperation, partnership and trust
The authors also implemented SOTA algorithms to test the evaluation suite
The key contribution is implementing the environment in JAX, which has a substantial improvement in rollout performance, directly speeding up training and evaluation of RL algorithms

**Strengths:**

1. The paper outlined the problem clearly
2. The authors presented a strong speed up improvement compared to the existing social dilemma environments
3. There is a diverse set of environments exploring many aspects of social dilemma
4. The authors provided a strong benchmark of RL algorithms for future comparisons

**Weaknesses:**

1. The authors clam to have a 50x+ speed up compared against the Melting pot 2.0. However, there were many differences in the observation space. It doesn't feel like a fair apples to apples comparison.
2. The effects of changing the layout grid-like observation was not discussed
3. It might be helpful to understand how the same RL algo compares between the SocialJax vs Melting Pot 2.0

**Questions:**

* Why was the observation space changed? Is it possible to have a similar observation space for comparisons (possibly as an option)

---

> ### Author Response · Authors · 2025-11-22
>
> We thank the reviewer for their insightful comments and constructive feedback. We appreciate the time and effort spent evaluating our work. We address the concerns below and provide clarifications, additional explanations of our paper.
>
>
> **Weakness 1: It doesn't feel like a fair apples to apples comparison.**
>
> > *The authors clam to have a 50x+ speed up compared against the Melting pot 2.0. However, there were many differences in the observation space. It doesn't feel like a fair apples to apples comparison.*
>
> We agree that Socialjax and Meltingpot are not exactly the same, which would cause the "apples to apples comparison" problem. We want to emphasize that our original goal was not to fully replicate MeltingPot, but rather to leverage its core design principles to build a training pipeline that is convenient for researchers. This allows researchers to quickly test and validate their ideas. Therefore, the core of our comparison is that we can enable very fast experimentation under agent interaction logic similar to MeltingPot, thereby providing practical support to the research community.
>
> **Weakness 2: The effects of changing the layout grid-like observation**
>
> > *The effects of changing the layout grid-like observation was not discussed*
>
> Transforming a pixel-based environment into a grid-based one simplifies the agent’s state space to some extent and reduces the burden of learning low-level perceptual features. However, we believe that social dilemma research is primarily concerned with whether agents learn the cooperative and competitive strategies. Whether a neural network can extract meaningful representations from raw pixel observations is not the main research goal of studying social dilemmas. We will explain the differences in the paper. It is an important change and we really should have mentioned it. Thank you.
>
>  **Weakness 3: The same RL algo compares between the SocialJax vs Melting Pot 2.0**
>
> > *It might be helpful to understand how the same RL algo compares between the SocialJax vs Melting Pot 2.0*
>
> Unfortunately, the A3C-family algorithms used in the MeltingPot 2.0 report are not open-sourced, and A3C is designed for multi-CPU asynchronous execution, making it impractical for us to reproduce the algorithms used in MeltingPot. Although the MeltingPot library includes a PPO implementation, it does not provide standard baselines. Therefore, to make a fair comparison with MeltingPot, we would need to re-implement the corresponding SocialJax algorithms within MeltingPot itself.
>
> In our experiments, running MeltingPot’s PPO on an A100 GPU and 14 cores CPU required 6–7 days per environment to reach the same number of timesteps as SocialJax in a few hours, highlighting the difficulty of reproducing MeltingPot’s results in practice. This challenge is one of the reasons we developed SocialJax. We want to provide the community with environments and benchmarks that make it much easier to validate new ideas and algorithms for solving social dilemmas.
>
> Finally, we use Schelling diagrams to show that our environments indeed exhibit social dilemma properties, supporting our position that SocialJax serves as a reliable and fast framework for social dilemma research.
>
>   **Questions 1: Why was the observation space changed**
>
> > *Why was the observation space changed? Is it possible to have a similar observation space for comparisons (possibly as an option)*
>
> JAX was originally designed to accelerate deep learning and provides rich parallelization interfaces, allowing us to run environments on GPUs. However, implementing environments in JAX comes with high development costs. Since we aim for the entire environment to be JIT-compilable on the GPU, we cannot use traditional `for` and `if` statements or rely on common dynamic variables like `list`. All code and environments logics must be structured with GPU parallelism. In principle, JAX can support pixel-based environments, but this would be extremely complex.
>
> Moreover, for social dilemma research, we consider the critical aspect to be the policy of cooperation and competition between agents. While pixel-based environments offer richer input, we believe that whether a neural network can interpret the pixel map is not the main focus for researchers validating their algorithms for solving social dilemmas.

---

> ### Author Response · Authors · 2025-11-28
>
> Dear Reviews,
>
> May we ask if your concerns have been addressed? We would be happy to answer any further questions and provide more details about our paper.
>
> Best Regards,
>
> Authors

---

### Official Review · Reviewer_HwDj · 2025-11-01

**Soundness:** 4
**Presentation:** 3
**Contribution:** 3
**Rating:** 6
**Confidence:** 5

**Summary:**

SocialJax presents a suite of Melting Pot–derived tasks implemented in a lightweight minigrid framework to probe sequential social dilemmas among multiple reinforcement-learning agents. Built on JAX, the benchmark emphasizes end-to-end vectorization to significantly speed up both training and evaluation. To validate each environment’s social incentives, if they encourage individualistic or altruistic behavior, the authors use Schelling diagrams produced via the SVO algorithm. They also evaluate several multi-agent variants of PPO, establishing standardized baselines and illustrating performance differences across tasks. The environments were evaluated to be different from each other as the number of cooperators-individual payoff curves differ between environments. The learning curves against PPO models are also quite different and thought provoking how these variations in PPO impact altruistic behaviour so much.

**Strengths:**

* Clear, well-structured writing that’s easy to follow.
* Learning-curve plots are clean and legible, making comparisons straightforward.
* Variations among PPO architectures and their returns provide informative contrasts.
* Effective use of JAX for MARL: aligns with non-stationarity in transitions, rewards, and observations, and the performance benefits are evident.
* Insightful comparison of individually rewarded PPO vs. common-rewarded PPO, plus PPO-RE’s exchange schedule.
* Well-chosen environments that focus on mixed-motive cooperation which is an important aspect of multi-agent interaction and underexplored.
* Performance descriptions and the accompanying Schelling analyses are reasonable and generally supported.
* The Melting-Pot suite is theoretically grounded in game theory and the inclusion to JAX is welcomed

**Weaknesses:**

* Most environments are direct ports from Melting Pot or prior work; Coins and STORM already have JAX implementations, so the contribution feels more like technical consolidation than novel insight into mixed-motive RL.
* All evaluated algorithms are PPO variants; including other cooperative MARL approaches would better test generality and strengthen the conclusions.
* The Schelling diagrams (via SVO) are under-explained and lack context; it’s unclear why they’re preferable to simple metrics like cumulative individual rewards (for individualistic agents) and cumulative common rewards (for altruistic agents) or maybe even behavioural metrics. Related discussion to Schelling and why the Schelling diagrams are better is important
* Agent partial-observation windows are not visualized in the environment images, limiting clarity about information constraints.
* The benchmark is restricted to Melting Pot–based environments, which may introduce intra-suite bias and limit external validity.
* The study does not examine mixed individual and common reward settings, omitting an important class of real-world mixed-motive scenarios.
* Lack of explanation for why PPO-RE struggles in coop mining leaves the analysis incomplete.
* Possible exploration deficits are not investigated; lack of exploration may explain lower returns for some PPO variants.
* No off-policy or value mixer baselines

**Questions:**

I enjoyed reading the paper and following the results. That said, I was left wanting a deeper understanding of what’s new beyond Melting Pot, since the suite is derived from it. SocialJax’s focus on mixed motives (individual vs. altruistic behavior) is valuable precisely because it avoids the extremes of full competition or full cooperation. With that in mind:

* One classic mixed competition–cooperation task is Level-Based Foraging (LBF, [https://github.com/semitable/lb-foraging](https://github.com/semitable/lb-foraging)), where agents compete for reward but sometimes benefit from cooperating. Would you consider expanding the suite to include LBF or similar non–Melting Pot tasks to mitigate intra-suite bias and strengthen generality?

* The authors compared purely individual vs. purely shared rewards, but not mixtures. Prior work (Wang et al., *Individual Reward Assisted Multi-Agent Reinforcement Learning*, : [https://proceedings.mlr.press/v162/wang22ao.html](https://proceedings.mlr.press/v162/wang22ao.html)) shows that adding individual rewards to shared objectives in SMAC can hurt performance; recent directions in a similar vain explore higher-gradient adjustments ([https://arxiv.org/abs/2505.20579](https://arxiv.org/abs/2505.20579)) and value-shaping heuristics ([https://arxiv.org/abs/2508.17696](https://arxiv.org/abs/2508.17696)). SocialJax seems well-suited to probe this mixed-motive effect. Could the authors evaluate the IRAT method from Wang et al. on their suite since it is PPO-based, and report how well it performs?

* Could the authors add a few baselines with modified versions of their environments that introduce individually rewarded “distractor” objects (e.g., apples) that disappear and pay out when stepped on? This should induce local individualistic behavior; measuring how agents balance these against common goals could deepen the robustness analysis of altruism for the different methods presented.

* A key missing baseline is from the family of opponent-shaping algorithms. Proximal Advantage Alignment (“adalign”) from *Advantage Alignment Algorithms* ([https://arxiv.org/abs/2406.14662](https://arxiv.org/abs/2406.14662)) is a PPO-style method that steers policies via advantage signals and reportedly succeeds on Commons Harvest in Melting Pot. Would you include adalign to broaden algorithmic diversity and test how it's coordination compares in your tasks?

---

> ### Author Response · Authors · 2025-11-22
>
> We sincerely thank the reviewer for their insightful comments and suggestions. Below, we address your questions, outline possible solutions, and provide some of the additional experiments you requested.
>
> **Weakness 1: Most environments are direct ports from Melting Pot or prior work**
>
> We acknowledge that our contribution is more like technical consolidation. However, we hope that our environments and algorithms provide tangible benefits to the community by improving the efficiency and accessibility of research on social dilemmas.
>
> Our motivation for building SocialJax comes directly from our experience studying social dilemmas. We found that, compared with many standard RL domains, existing social dilemma environments are less accessible and more difficult to experiment with. We hope that SocialJax will serve as a practical and reliable platform for researchers, enabling faster experimentation and broader exploration of cooperative and competitive behaviors in social dilemmas.
>
> **Weakness 2: All evaluated algorithms are PPO variants**
>
> We also include VDN as value-based baselines, and our experiments on Harvest: Open produce reasonable and consistent results. The algorithm implementations can be found at: https://anonymous.4open.science/r/SocialJax-23F3/algorithms/. The results of Harvest: Open can be found (https://anonymous.4open.science/r/SocialJax-23F3/tests/vdn_results/.)
>
> **Weakness 3: Schelling diagrams**
>
> The Schelling diagram provides a detailed characterization of how  interactions between agents and environments change as the numbers of cooperators and defectors vary. It reveals how agents following different strategies interact under mixed populations. For example, in Harvest:Open, we observe that when even a single defector is present, the cooperators’ reward drops sharply. This indicates that one defector can consume nearly all the apples, collapsing the ecological balance of the environment. In the Clean Up, the reward increases significantly only when the number of other cooperators exceeds 2. This is because a sufficient number of cooperative agents is required to clean the river; without enough cooperators maintaining the environment, apples cannot grow.
>
> **Weakness 4: partial-observation windows**
>
> We can address this concern. In fact, our code already includes this function, and the agents’ fields of view are visualized in the GIFs available in our anonymous GitHub repository.
>
> **Weakness 5: The benchmark is restricted to Melting Pot–based environments**
>
> We believe that MeltingPot currently provides some of the most comprehensive and challenging environments for studying social dilemmas, allowing us to cover a wide range of research scenarios. Following your suggestion, We will consider implementing LBF environments during the review period.
>
> **Weakness 6: Mixed individual and common reward settings**
>
> Our Mushroom environment is a mixed-motive environment. Agents encounter different types of mushrooms, each representing distinct social incentives. Red mushrooms represent selfish behavior, providing a reward of +1 only to the consuming agent and having the shortest digestion time (agent cannot move during digestion time which is a punishment can stops them collect mushrooms). Green mushrooms provide a reward of 2 shared equally among all players, reflecting cooperative behavior that benefits both the consumer and others. Blue mushrooms (which cause the longest digestion time) provide a reward of 3 shared among all players except the consumer, representing altruistic behavior that sacrifices individual benefit for the group.
>
> **Weakness 7: Lack of explanation for why PPO-RE struggles in coop mining**
>
> Because PPO-RE needs to sweep over preference-related parameters during training to adjust the balance between selfish and altruistic behaviors, a single PPO-RE run naturally requires more timesteps. Due to the limited number of timesteps, it is possible for the agent’s policy to become stuck in a selfish local optimum. Besides, in Coop Mining there is no explicit penalty for selfish behavior, and a selfish agent can still obtain reasonable rewards. This may explain the lower returns observed for PPO-RE in experiments.
>
> **Weakness 8: Possible exploration deficits are not investigated**
>
> Your point is valid, especially for PPO-RE, which requires adjusting parameters during training to balance selfish and altruistic behaviors which make it more time steps. We have made our best effort to ensure fair comparisons across algorithms. In fact, other methods specifically designed for social dilemmas also require repeated runs to search for suitable hyperparameters (for example, selecting the SVO angle or tuning the beta coefficient in Advantage Alignment Algorithms). We are considering increasing the training time for PPO-RE and providing a clearer explanation of how PPO-RE works.
>
> **Weakness 9: No off-policy or value mixer baselines**
>
> Seeing Weakness 2, We add VDN algorithm.

---

> > ### Author Response · Authors · 2025-11-22
> >
> > **Question 1: Level-Based Foraging**
> >
> > See Weakness 5. We will consider implementing your suggestions during the review period.
> >
> > **Question 2: IRAT method**
> >
> > In SVO and PPO-RE, agents receive different proportions of individual and shared rewards under different hyperparameters. Therefore, the algorithms already take mixed rewards during training.
> >
> > **Question 3: Individually rewarded “distractor” objects**
> >
> > Seeing Weakness 6, in the mushroom environments, We believe the red mushroom already serves as an individually rewarded distractor object, which only benefits the consuming agent.
> >
> > **Question 4: A key missing baseline is from the family of opponent-shaping algorithms**
> >
> > Beyond the *Advantage Alignment Algorithms*, this also includes the IRAT algorithm and the LBF environment. We agree that these experiments would strengthen the paper, and we will consider which results are most feasible to include within the review timeline.
> >
> > Finally, we sincerely thank the reviewer for their valuable suggestions. Because the experimental scope is large, we will first test the newly added environments and algorithms in a small subset of scenarios. Once the results are stable and we have verified that no implementation issues and coding bugs remain, we will extend the evaluation to all environments.

---

> > > ### Comment · Reviewer_HwDj · 2025-11-25
> > >
> > > Dear Authors,
> > >
> > > Thank you for implementing and evaluating VDN on the Harvest environment. I think this baseline and implementation improves SocialJAX's algorithmic diversity and will be beneficial in future work. I also noticed that there is a new plot showing VDN's return is above MAPPO's almost at the level of IPPO with common rewards. Would the authors be able to add any insights or analysis on performance of VDN, if time permits, relative to that of PPO-RE and SVO?
> > >
> > > Additionally, the implementation of Level Based Foraging has expanded the suite from being just a revamped Melting Pot. I also appreciate implementation of IRAT which appears to underperform against IQL if their environments are the same. I find SocialJAX and MeltingPot are interesting due to being grounded in game theoretic scenarios. Methods like IRAT are designed without these scenarios in mind but appear to be able to handle social dilemmas.
> > >
> > > I am satisfied with the other responses and  will increase my current rating.

---

> ### Author Response · Authors · 2025-11-28
>
> Dear Reviewer,
>
> We greatly appreciate the reviewer’s recognition and valuable feedback.
>
> In the Harvest Open environment, each agent can achieve very good returns by performing similar local behaviors (like harvesting apples according to their respawn frequency). With team reward, each IPPO agent only needs to adjust its harvesting rate based on nearby apples and agents, which makes learning and convergence easy. However, MAPPO must learn a centralized critic over the joint space, resulting in a higher dimensional network, slower learning, and unstable credit assignment. As for VDN, although linearly decomposing the team Q-value cannot fully represent the fundamentally nonlinear cooperative dynamics (e.g., two agents harvesting simultaneously may reduce long-term apple regeneration). It can still learn reasonably good strategies when only one agents harvesting the patch of apples.
>
> When we examine the training curves, we find that MAPPO performs best in the Clean Up environment. We attribute this to the fact that, in Clean Up, achieving optimal performance requires agents to assume different functional roles. Ideally, some agents continuously clean the river at the bottom of the map, while others focus on collecting apples at the top. Because if an agent finishes cleaning the river and then moves across the map to the other side to collect apples, it wastes time and cannot achieve maximum efficiency. MAPPO is better suited to handle such tasks because its centralized critic can more effectively perform agent credit assignment for these differentiated roles.
>
> PPO-RE is inherently more tricky, because it requires sweeping hyperparameters during training. In contrast, for algorithms like SVO, one can simply sweep a hyperparameter, retrain with the new setting, and continue until better hyperparameters are found. If we ensure the same number of timesteps for all algorithms, this actually puts PPO-RE at a disadvantage compared to SVO. Therefore, we believe that this point may need to be clarified.
>
> We sincerely thank you and would be happy to address any further questions.

---

### Author Response · Authors · 2025-12-01

Dear AC,

We sincerely thank the AC and the reviewers for their time, effort, and valuable feedback. Thank you very much to the AC for the additional effort.  We yould like to provide a summary of reviews' comments and our rebuttal to help you faster understand our progress.

Reviewer A4zv gave us a score of 8. They raised concerns regarding **the gap between our pixel-level environment and the grid-level environment**, **the performance differences of the same algorithms on SocialJax and MeltingPot**, and **the explaining of the Schelling diagram in the Gift environment**. We addressed these concerns by offering more explanation and received positive feedback, and the reviewer maintained the score of 8.

Reviewer HwDj initially gave us a score of 6 and raised concerns about **the novelty of our environments**, **the Schelling diagram**, and **the mixed-reward setting**. We provided detailed explanations, and most of these issues were already addressed in our paper and code. Reviewer HwDj also requested additional experiments on new environments such as LBF, VDN, IQL, IRAT, and part of AAA algorithms. We basically completed these requests and improved **the reviewer’s score from 6 to 8**, a few days before the API leaking bug.

Reviewer RUpe gave us a score of 6. Reviewer RUpe’s comments primarily concern the **differences between our SocialJax and MeltingPot**, specifically focusing on the distinction between our grid-level environment and MeltingPot’s pixel-level environment, as well as the resulting implications. We have provided a detailed explanation addressing these points, with the goal of leveraging MeltingPot’s core design principles to build efficient and user-friendly environments and algorithms. It is worth noting that **Reviewer A4zv** raised similar concerns regarding the impact of  SocialJax’s grid-level implement, and expressed satisfaction with our response.

Reviewer m7Ji gave us score of 4, but we believe the concerns of our link is not a real issue. Reviewer m7Ji raised concerns regarding the novelty of our environment and the accessibility of our code repository, specifically questioning whether our implementation has been open sourced.  We would like to clarify that the anonymous github link we provided has been verified and confirmed to be accessible (other reviewers have successfully accessed it). Our code is indeed open sourced and released under the Apache License 2.0. We provided a detailed explanation along with additional engineering descriptions of our work. Regarding the novelty of our environment, this concern aligns closely with that raised by **Reviewer HwDj**. We addressed both reviewers’ questions with the similar technical clarification, and Reviewer HwDj expressed satisfaction with our response.


Reviewer CDVN assigned our paper a score of 2, but we believe this evaluation stems largely from a misunderstanding of our work. Our contribution is implementing both environments and agent algorithms in JAX, enabling **fully end-to-end GPU-based training**. We verify social dilemma properties using Schelling diagrams and provide benchmarks across algorithms. ***Our implementation is written from scratch in JAX and does not reuse nor depend on MeltingPot code.*** However, **Reviewer CDVN appears to have interpreted our work as merely running JAX agents on existing MeltingPot CPU-based environments**. This is
evident from their question: "*Did you use the same GPU cores as Agapiou et al.? Are the performance figures taken from that paper?*", which suggests they believe our results were taken from MeltingPot.  This is actually a complete misunderstanding of our paper. We only drew inspiration from MeltingPot’s core design principles to build our own environments and algorithms and make them fully run on GPUs. Almost all other stated weaknesses also stem from the same misunderstanding, which include **CPU-based Parallelism**, **Model implementation**, **CPU performance**, etc.

In summary, we have addressed the similar concerns raised by different reviewers and provided the additional experiments requested (we add LBF environments, IQL, IRAT, VDN, and part of AAA in our anonymous github), which makes Reviewer HwDj improved **the score from 6 to 8**. We have also responded accordingly to other issues, such as questions about our code link and **misunderstandings** (Reviewer CDVN seems like believing that our work merely adapts agents to run efficiently in JAX on GPUs and uses the original CPU-based MeltingPot environments.), which is **incorrect**.  We have built a fully GPU-based environment and algorithm suite using JAX, with no code reuse or dependency on MeltingPot.

We sincerely appreciate the AC’s efforts and support, and we hope this summary will assist the AC in evaluating our submission.

Best Regards,

Authors

---

### Meta-Review · Area_Chair_SngY · 2026-01-07

**Summary:**

The paper proposes a JAX suite for sequential social dilemma environments. They also add baseline methods in multi-agent reinforcement learning.

The authors did a good job of addressing the reviewers, and the consensus is to accept the paper.

**Reviewer Concerns:**

Most reviewers recommend acceptance. Reviewer CDVN was sceptical, but the authors provided a detailed response.

**Reviewer Scores:**

m7Ji will increase their score

---

### Decision · Program_Chairs · 2026-01-26

Accept (Poster)